# GTalign: spatial index-driven protein structure alignment, superposition, and search

Mindaugas Margelevičius [ORCID] [1] [✉]

With protein databases growing rapidly due to advances in structural and computational biology, the ability to accurately align and rapidly search protein structures has become essential for biological research. In response to the challenge posed by vast protein structure repositories, GTalign offers an innovative solution to protein structure alignment and search—an algorithm that achieves optimal superposition at high speeds. Through the design and implementation of spatial structure indexing, GTalign parallelizes all stages of superposition search across residues and protein structure pairs, yielding rapid identification of optimal superpositions. Rigorous evaluation across diverse datasets reveals GTalign as the most accurate among structure aligners while presenting orders of magnitude in speedup at state-of-the-art accuracy. GTalign's high speed and accuracy make it useful for numerous applications, including functional inference, evolutionary analyses, protein design, and drug discovery, contributing to advancing understanding of protein structure and function.

In contemporary structural bioinformatics, the advent of advanced artificial neural network architectures[1–3] has ushered in an era where protein structures are predicted with high accuracy for a myriad of protein sequences[4,5]. This surge in structural data has presented a challenge: the need for efficient and rapid protein structure comparison to distill meaningful insights from the burgeoning repositories of three-dimensional protein structures. Such tools enable the extraction of biologically relevant information, decipher evolutionary relationships[6–8], and contribute significantly to understanding functional mechanisms encoded within protein structures[9–11]. In this context, the development of computational tools for rapid large-scale protein structure alignments represents an important step forward.

The need for efficient tools gave rise to a variety of computational techniques, each with its own strengths and limitations, aiming to achieve accurate and rapid comparisons of protein structures. Two fundamental approaches have emerged: local pattern matching and rigid-body superposition optimization. Local pattern matching involves analyzing structures independently, making it suitable for handling flexible regions like linkers between protein domains. This approach encompasses various strategies such as optimizing the match between protein inter-residue distance matrices[12] or probability distributions[13], aligning secondary structure elements using double dynamic programming[14] or graph matching[15], finding and extending[16] or chaining[17] aligned fragment pairs with optimal inter-residue distance matching, and quantifying evolutionary similarity to infer initial alignments[13,18].

On the other hand, methods based on rigid-body superposition treat structures as rigid bodies and focus on optimizing their local or global spatial agreement. These methods may involve optimizing superposition for protein fragments[19], iterative superposition at different distance cutoffs[20], or multi-stage fragment-based superposition optimization[21]. To achieve accurate pairwise protein alignments, these methods[20,21], and those employing local pattern matching[14–18], apply iterative alignment refinement that includes rigid-body superposition.

While approaches like TM-align[21] and Dali[22] have been essential in advancing protein structure comparison, their computational complexity has made them less practical for large-scale applications to growing protein structure databases. To address this challenge, some strategies involve preprocessing database structures to compute

[1]Institute of Biotechnology, Life Sciences Center, Vilnius University, Vilnius, Lithuania. [✉]e-mail: mindaugas.margelevicius@bti.vu.lt

computationally intensive parts in advance[15]. Alternatively, accelerating protein structure searches involves conducting searches across preclustered databases[23,24]. However, these approaches require compute-intensive preprocessing and constant updating.

Recently, Foldseek[25] has introduced an approach of leveraging a structural alphabet to reduce the problem to a sequence comparison and employed k-mer matching to filter out insignificant matches. Despite its notable speed, Foldseek contends with inherent limitations in alignment accuracy, underscoring the persistent challenge of achieving a balance between computational efficiency and accuracy.

We bridge this gap by introducing GTalign (Giga-scale Targeted alignment), a computational tool designed for extensive protein structure similarity exploration and alignment. GTalign's primary objective is to find an optimal spatial overlay for a given pair of structures and subsequently derive an alignment from it. In that sense, GTalign treats each structure as a rigid body and evaluates protein similarity through the identification of the best superposition, thereby ensuring an accurate structural alignment. High computational performance is achieved by algorithms developed for low-complexity superposition search and to harness the power of modern computer processors and accelerators. Consequently, GTalign combines high accuracy and fast execution times, offering a solution for accelerated and accurate structural analyses.

## Results and discussion
### GTalign approach
For a given protein pair, GTalign employs an iterative process that involves (i) selecting a subset of atom pairs, (ii) calculating the transformation matrix, (iii) deriving an alignment based on the resulting superposition, and finally, selecting the alignment that maximizes the TM-score[26]—a strategy akin to TM-align[21]. The first two steps can be naturally parallelized for speed. However, alignment derivation entails finding, for each residue, the spatially closest residue of the other structure after superposition. Preserving sequence order is crucial for assessing protein topology similarity, and thus, this optimization typically requires dynamic programming (DP), a standard choice for solving such problems. Handling positional dependencies while maintaining sequence order poses challenges to DP implementation with favorable time complexity. Therefore, this, coupled with the need for multiple alignments per structure pair, represents a bottleneck for large-scale computation despite optimized parallel DP implementation efforts[27].

GTalign tackles this challenge by introducing a spatial index for each structure, which allows for considering atoms independently and ensures $O(1)$ time complexity for the alignment problem (Fig. 1a). Although post-processing is necessary to preserve sequence order, it has sublinear rather than quadratic time complexity. This methodology enables GTalign to effectively parallelize all steps, efficiently navigating through an extensive superposition space. Coupled with parallel processing of numerous protein structure pairs, it significantly accelerates the entire protein similarity search process.

### Comprehensive reference-free performance evaluation
We benchmarked GTalign against a set of well-established protein structure aligners: TM-align[21], Dali[22], FATCAT[28], DeepAlign[18], and Foldseek[25]. We also evaluated the performance of TM-align's fast variant (option `-fast`) and Foldseek variants utilizing both fast (`--tmalign-fast 1`; FoldseekTM) and regular (`--tmalign-fast 0`) versions of TM-align for aligning protein structures that passed sequence similarity filters. Various parameterizations of GTalign were also benchmarked. The `--speed` option controls the algorithm's execution speed, with higher values prioritizing speed over accuracy. In addition, the `--pre-similarity` and `--pre-score` options were employed for initial similarity screening in the sequence and structure space, respectively.

Comprehensive evaluations were conducted across four diverse datasets representing different protein analysis scenarios: (i) SCOPe 2.08[29] protein domains filtered to 40% sequence identity, (ii) PDB[30] full-length structures filtered to 20% sequence identity, (iii) the UniProtKB/Swiss-Prot[31] protein structures from the AlphaFold Database[4], and (iv) the HOMSTRAD database[32].

The evaluation employed an unbiased and reference-free approach that reveals alignment accuracy through the superposition of structures based on the produced alignments (see the section Alignment accuracy evaluation). We calculated the superposition score, TM-score[26], for alignments produced by each tool. TM-score ranges from 0 to 1, with 1 representing a perfect match. Therefore, the progression of TM-scores serves as an indicator of a tool's accuracy. The range TM-score $\in [0.5; 1]$ is particularly important because it reflects sensitivity, i.e., a tool's ability to detect proteins sharing the same fold[33].

TM-score can be normalized by the length of either of the two proteins being compared. We first discuss the results obtained using the TM-score normalized by the length of the shorter protein. This TM-score evaluates structural similarity irrespective of the ratio of the two protein lengths, effectively capturing similarities of smaller proteins or domains matching regions of larger proteins.

The results (Fig. 1b–d, Supplementary Section S1.1, Supplementary Fig. S1a, Supplementary Table S1) show that GTalign consistently outperforms all the aligners in terms of accuracy. GTalign (option `--speed=0`) produces up to 7% more alignments with a TM-score ≥ 0.5 than TM-align, the second most accurate tool (732,024 vs. 683,996, SCOPe40 2.08 dataset).

This trend persists across the entire TM-score significance range from 0.5 to 1.0 (Supplementary Table S2). However, there are exceptions: GTalign reports 191 fewer alignments than TM-align for TM-scores ≥0.7 in the PDB20 dataset and 31 fewer alignments for TM-scores ≥0.8 in the Swiss-Prot dataset. Further investigation revealed that these differences stem from aligning queries with short proteins with fewer than 30 residues. For a pair of proteins, GTalign employs approximate partial sorting to select candidate alignments for detailed refinement. Alignments for very small proteins or peptides score similarly, and this approximation can occasionally lead to a suboptimal final alignment.

While the issue of short proteins is recognized and left for future resolution, analysis based on the TM-score normalized by the query length effectively diminishes the significance of alignments between queries and much smaller proteins. In this evaluation setting (Fig. 2, Supplementary Fig. S1b, Supplementary Table S3), GTalign demonstrates superior accuracy over the other aligners again. For example, in the SCOPe40 2.08 dataset, GTalign (option `--speed=0`) produces up to 7% more alignments with a TM-score ≥0.5 than TM-align (492,887 vs. 460,847). Disregarding insignificant differences of one alignment, GTalign exhibits superiority across the full TM-score significance range (Supplementary Table S4).

GTalign also demonstrates higher accuracy on the HOMSTRAD dataset (Fig. 3), which contains reference structure alignments grouped into evolutionarily and structurally related protein families. These families exhibit relatively high structural similarity, as evidenced by the TM-score distribution of the reference alignments (Fig. 3). Therefore, these results highlight GTalign's utility in improving structural alignments, even among highly similar proteins. This is further supported by the greater accuracy of GTalign-produced alignments compared to the reference alignments, suggesting potential applications in constructing reference datasets and classifying proteins. A similar trend was observed previously[34].

In addition, we provide structural examples in Supplementary Figs. S2–S5, each corresponding to one dataset used in our benchmark study. These examples highlight nontrivial structural similarities identified by GTalign but overlooked or misaligned by all other

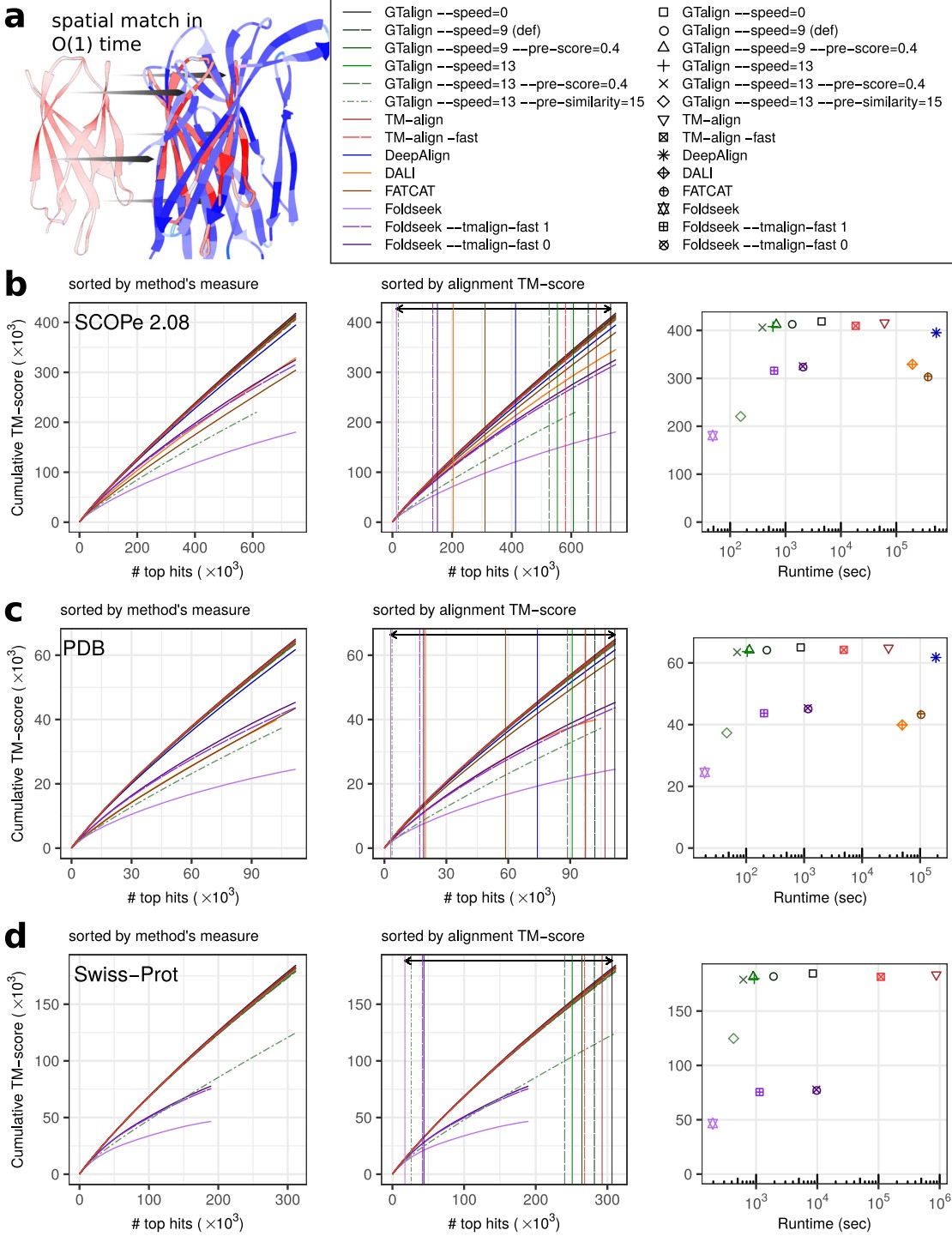

**Fig. 1 | Results. a** Illustration of matching protein structures with $O(1)$ time complexity. GTalign explores numerous superpositions in parallel. Upon obtaining a superposition, the alignment between the query protein (red) and the subject protein (blue) is generated using the subject protein's spatial index. This index allows for the independent retrieval of the nearest residue in the subject protein for each residue in the query protein, enabling parallel processing. **b** Benchmarking results on the SCOPe40 2.08 dataset with 2045 queries and 15,177 database entries. Parameterized runs of GTalign, TM-align, and Foldseek are included. The left panel plots the cumulative TM-score (normalized by the shorter protein length) against the number of top alignments ranked by a tool's measure (TM-score, Z-score, or P-value). In the middle panel, the alignments are sorted by their (TM-align-obtained) TM-score. Vertical lines indicate the number of alignments with a TM-score ≥0.5. The arrow denotes the largest difference in that number between GTalign (732,024) and Foldseek (13,371). The right panel shows the cumulative TM-score plotted against runtime in seconds. **c** Benchmarking results on the PDB20 dataset with 186 queries and 18,801 database entries. **d** Benchmarking results on the Swiss-Prot dataset with 40 queries and 542,378 database entries. DeepAlign, Dali, and FATCAT are excluded due to their long projected execution times. The Foldseek curves appear truncated due to the total number of hits it produced. The axes scales in Panels **b**–**d** are chosen to accommodate the maximum values of the cumulative TM-score and the number of top hits with a TM-score ≥0.5. Source data are provided as a Source data file.

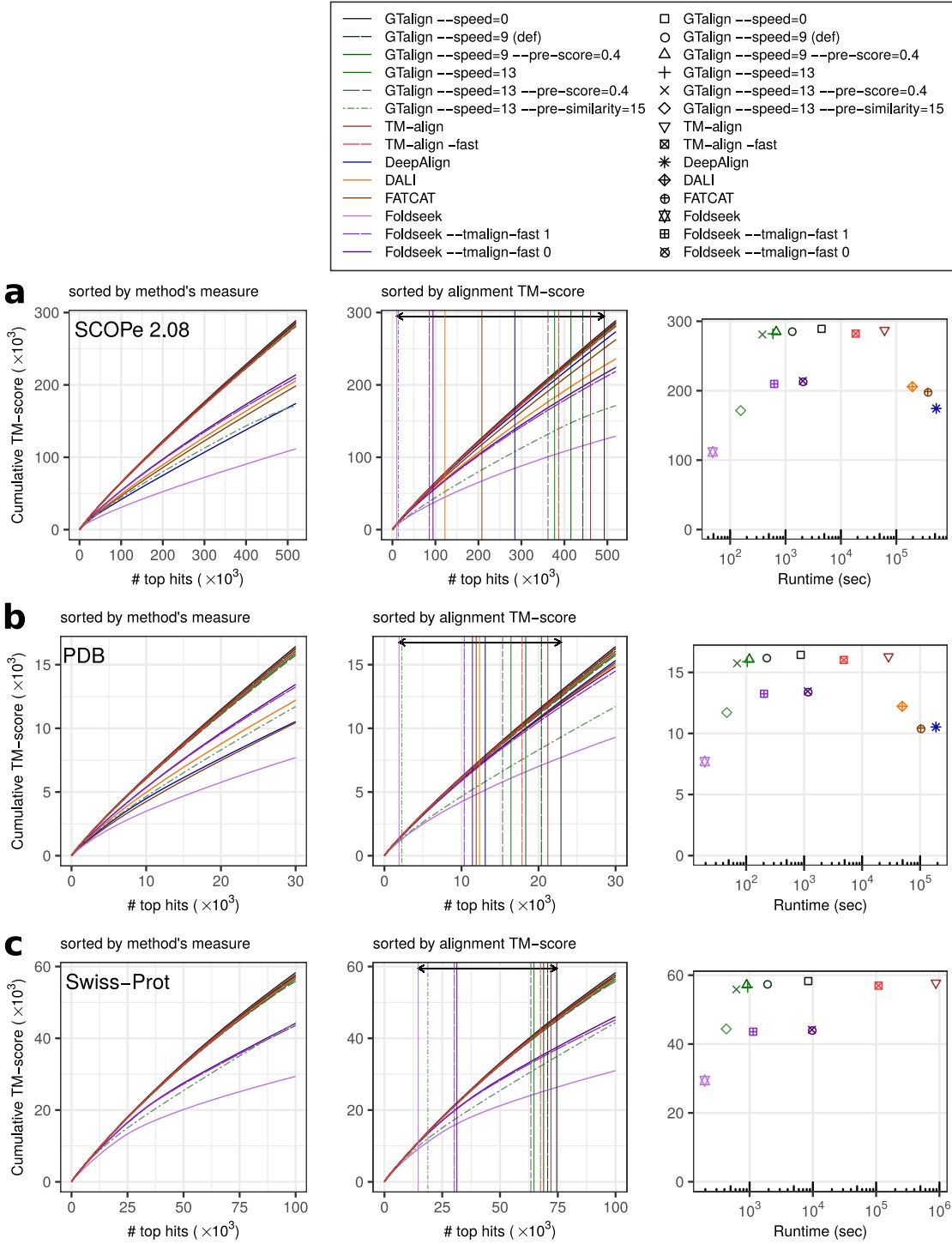

**Fig. 2 | Alignment evaluation results based on the TM-score normalized by the query length. a** Benchmarking results on the SCOPe40 2.08 dataset with 2045 queries and 15,177 database entries. The left panel plots the cumulative TM-score (normalized by the query length) against the number of top alignments ranked by a tool's measure (TM-score, Z-score, or P-value). In the middle panel, the alignments are sorted by the TM-align-obtained TM-score. Vertical lines denote the number of alignments with a TM-score ≥0.5. The arrow highlights the largest difference in that number between GTalign (492,887) and Foldseek (10,375). The right panel shows the cumulative TM-score plotted against runtime in seconds. **b** Results on the PDB20 dataset with 186 queries and 18,801 database entries. **c** Results on the Swiss-Prot dataset with 40 queries and 542,378 database entries. Excluding DeepAlign, Dali, and FATCAT due to their long projected execution times. Source data are provided as a Source data file.

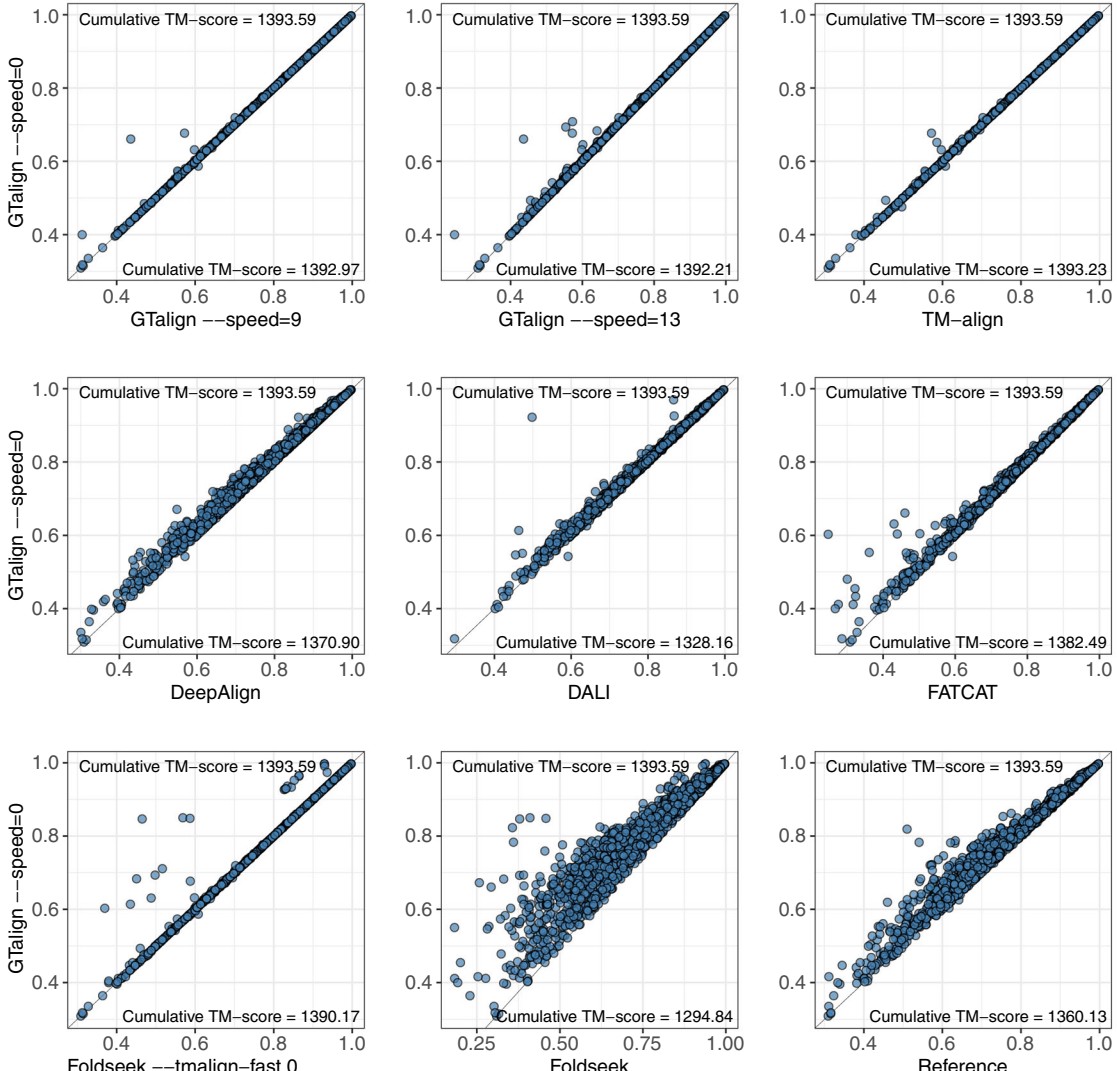

**Fig. 3 | Benchmarking results on the HOMSTRAD dataset.** Each plot illustrates the distribution of TM-scores between two tools. The black line indicates identical distributions. 100 structure pairs for which Dali did not produce any alignment are excluded from the corresponding plot. "Reference" represents the original alignments in the database. Source data are provided as a Source data file.

benchmarked tools. Notably, all showcased examples demonstrate domains (even from different folds; see a subsection below) or larger significant structural segments, with insertions and deletions, sharing the same topology, as confirmed by the TM-scores and structural alignments. Therefore, Supplementary Figs. S2–S5 exemplify GTalign's primary objective of achieving optimal protein spatial superpositions and detecting subtle yet significant structural similarities.

While GTalign excels in uncovering structural similarities, its efficiency also sets it apart (Fig. 1b–d). GTalign is up to 104–1424x faster (618–8454 vs. 879,965 s, Swiss-Prot dataset) than TM-align parallelized on all 40 CPU threads. It achieves a 177x speedup (options `--speed=13 --pre-score=0.4`) over the fast (`-fast`) TM-align version (618 vs. 109,319 s, Swiss-Prot dataset) and is the fastest among the tools except Foldseek.

Clearly, the sequence prefiltering strategy contributes to Foldseek's high speed. However, this comes at a high price in accuracy and sensitivity (only 13,371 alignments with a TM-score ≥ 0.5; SCOPe40 2.08 dataset, Fig. 1b; see also Fig. 2 and Supplementary Tables S2 and S4). When GTalign is configured to use sequence prefiltering (options `--speed=13 --pre-similarity=15`), a similar pattern emerges, with runtimes comparable to Foldseek (428 vs. 196 s, Swiss-Prot dataset) but decreased sensitivity. In contrast, no such effect is observed for prescreening in the structure space (option `--pre-score`; Figs. 1 and 2).

This phenomenon can be attributed to at least two factors. First, low sequence similarity does not necessarily correlate with low structural similarity, as demonstrated by the results. Second, the generation of accurate structural alignments using a single scoring scheme per protein pair may lack consistency. To address this, we explored the use of scores derived from spectral analysis of rotation-invariant two-dimensional representations of geometric features, such as angles and distances between residues, in the frequency (Fourier) domain. However, this approach demonstrated inconsistent results and requires further investigation. Despite these observations, leveraging prescreening in the sequence space can prove valuable for high-similarity searches.

GTalign offers additional computational advantages by providing the option to utilize multiple GPUs for computation. This feature was effectively leveraged for processing the SCOPe40 2.08, PDB20, and Swiss-Prot datasets, where GTalign exploited the computational power

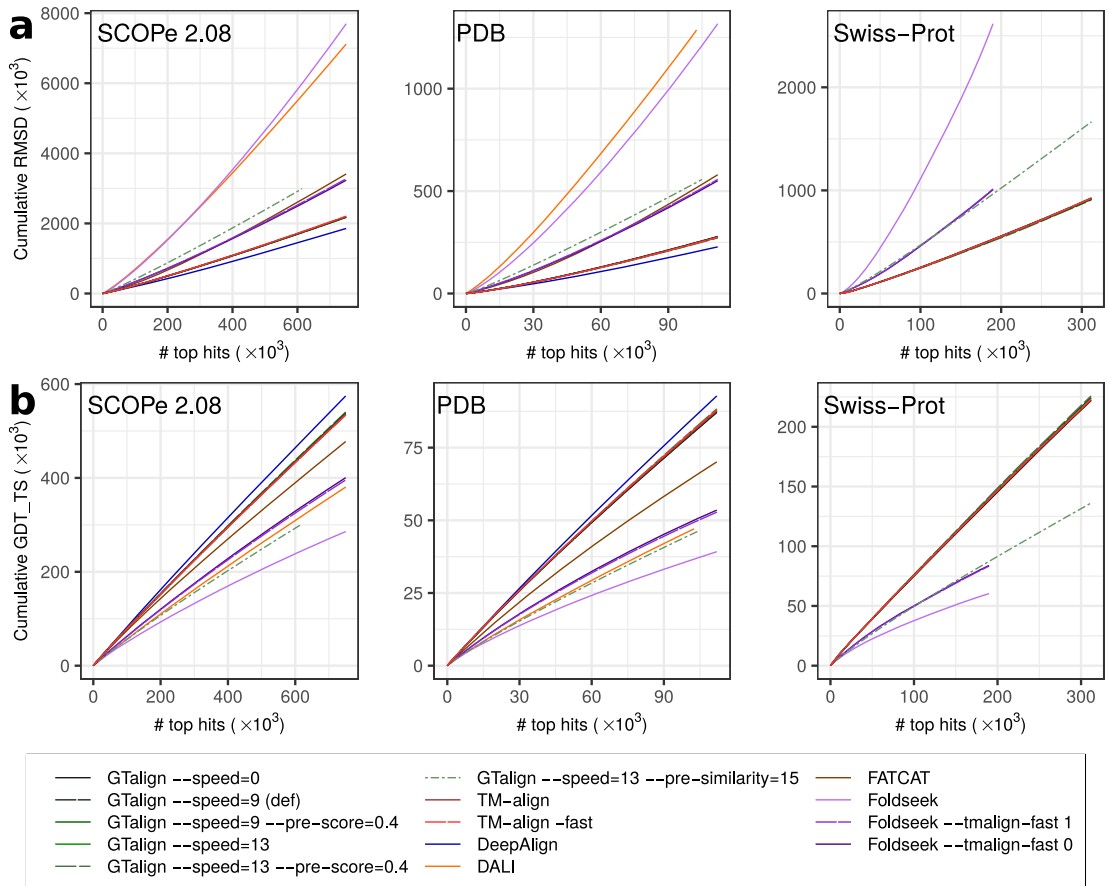

**Fig. 4 | Results of local alignment evaluation.** Alignment evaluation results for the SCOPe40 2.08, PDB20, and Swiss-Prot datasets using RMSD (root-mean-squared deviation, **a**) and GDT_TS (global distance test) scores (**b**) normalized by the number of aligned residue pairs. The panels display the cumulative RMSD and GDT_TS scores plotted against the number of top alignments ranked by a tool's measure (TM-score, Z-score, or *P*-value). Foldseek alignments were ranked by Foldseek's TM-score, resulting in lower RMSDs and higher GDT_TS scores. The curves of the TM-align and GTalign variants, excluding GTalign `--speed=13--pre-similarity=15`, closely match. Source data are provided as a Source data file.

of all three Tesla V100 GPUs available on the system. Supplementary Table S1 provides GTalign runtimes on one, two, and three GPUs, demonstrating scalability across all benchmarked parametrizations. Furthermore, the results presented in Supplementary Section S1.2 and Supplementary Table S5 unveil a noteworthy performance trend: A more recent desktop-grade GPU consistently outperforms the computational capabilities of the three server-grade V100 GPUs, effectively conveying GTalign's remarkable performance even when run on a single, relatively inexpensive GPU.

**Alignment accuracy evaluation using RMSD and GDT_TS**

TM-score, used to evaluate alignment accuracy in the previous subsection, is a global measure sensitive to alignment coverage due to normalization by protein length. Here, we turn to root-mean-squared deviation (RMSD) and global distance test (GDT)[20] for evaluation.

RMSD, a measure of spatial proximity, is normalized by the number of aligned residue pairs and is effective at capturing accurately aligned local protein regions (see Supplementary Fig. S6 in Supplementary Section S1.3 for an example). However, optimizing alignments based solely on RMSD can yield short aligned fragments that provide limited insight into structural similarity at the domain or protein level. Previous approaches[14,15,20] sought to find a balance between RMSD and alignment coverage to generate alignments sufficiently long to assess structural similarity without being overly divergent.

The GDT score (GDT_TS)[20] is another measure of spatial proximity, calculated at four different distance thresholds (1, 2, 4, and 8 Å), which does not over-penalize spatially unmatched residue pairs.

In this section, our RMSD and GDT_TS-based evaluation focuses on local alignments within alignment boundaries, providing insight into the extent to which alignments can be shortened to increase local precision by reducing alignment coverage. The results (Fig. 4) reveal that even among local alignment methods such as Foldseek (default parametrization), FATCAT, and DeepAlign, only DeepAlign produces alignments with lower RMSDs and higher GDT_TS scores than GTalign (Supplementary Table S6). However, DeepAlign achieves 15% and 18% lower RMSDs (2.48 and 2.03 vs. 2.91 and 2.47 for the SCOPe40 2.08 and PDB20 datasets) with 20% and 17% fewer aligned residues on average (56.0 and 67.4 vs. 47.6 and 57.1) compared to GTalign (`--speed=0`). The average TM-scores of GTalign alignments (Supplementary Table S1) exceeding those of DeepAlign alignments suggest that the difference in the number of aligned residues does not imply misaligned pairs. Indeed, Supplementary Table S1 and Supplementary Table S6 demonstrate that both GTalign and TM-align produce alignments that strike a good balance between coverage and precision. On average, GTalign (`--speed=0`) achieves lower RMSDs with higher alignment coverage compared to TM-align. In some cases, this is achieved by identifying different, more optimal spatial superpositions (see Supplementary Figs. S2–S4 for examples).

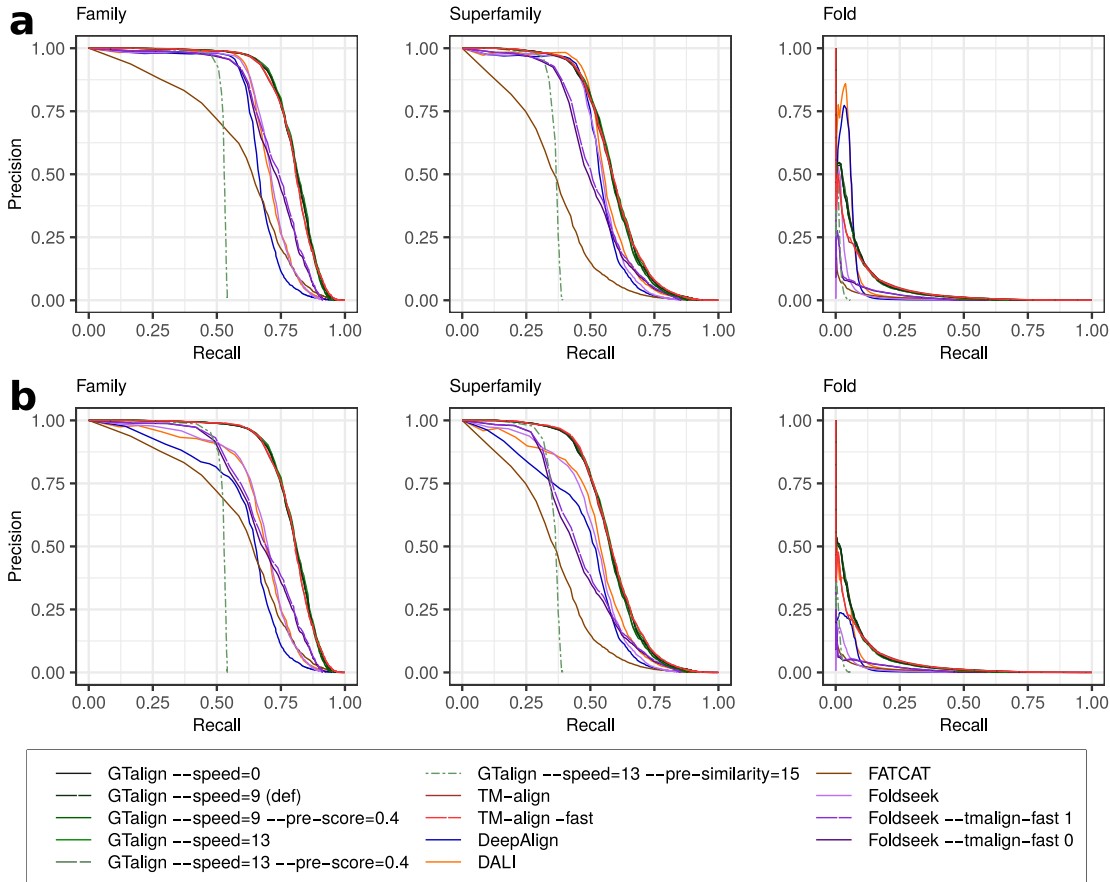

**Fig. 5 | Weighted precision-recall (PR) curves at the family, superfamily, and fold levels.** The areas under these PR curves are reported in Supplementary Table S7. **a** False positives for calculating precision and recall are pairs of structures from different SCOPe 2.08 folds, with the exception that those pairs belonging to Rossman-like (c.2–c.5, c.27, c.28, c.30, and c.31) or beta-propeller (b.66–b.70) folds are ignored[35]. **b** False positives correspond to pairs from different SCOPe 2.08 folds without exceptions. Source data are provided as a Source data file.

## Benchmarking against the SCOPe dataset reference

GTalign's main objective is to achieve an optimal superposition for a pair of structures for inferring their structural similarity, not rarely indicating an evolutionary relationship. This section assesses GTalign and the other tools from an evolutionary standpoint by examining their ability to replicate SCOPe classification. Importantly, this benchmark does not measure alignment accuracy or the rate of accurate alignments (Figs. 1–3) but rather the consistency between a tool's ranking of structure pairs and SCOPe classification.

The SCOPe knowledgebase categorizes protein domains into families, superfamilies, folds, and classes. Families group domains based on sequence similarity, with those sharing a common ancestor organized into superfamilies. Folds comprise structurally similar superfamilies, while classes are arranged by secondary structure content and organization[29].

Figure 5 shows the relationship between the precision and recall (PR) of matching domains of the same SCOPe 2.08 family, superfamily, and fold for each tool. As discussed in the next subsection, significant structural similarities extend even across folds and classes. Consequently, Fig. 5a displays the PR curves obtained with false positives (FPs) as pairs of domains from different folds, excluding well-known cross-fold relationships across Rossman-like (c.2–c.5, c.27, c.28, and c.31) and beta-propeller (b.66–b.70) folds[35]. In addition, Fig. 5b shows the PR curves when disregarding cross-fold relationships, with FPs corresponding to pairs from different SCOPe 2.08 folds.

The results in Fig. 5 and the areas under the PR curves (AUPRCs) reported in Supplementary Table S7 (Supplementary Section S1.4)

demonstrate that GTalign (variants `--speed=0` and `--speed=13`) generally outperforms the other tools, except for Dali in the evaluation that ignores cross-fold relationships at the fold level, where the difference in AUPRC is <1%. When evaluating sensitivity in identifying related domains before encountering the first FP (Supplementary Fig. S7), GTalign shows lower average sensitivity compared to both TM-align versions at the family and fold levels, and compared to Dali and DeepAlign at the superfamily level. However, these differences in the distributions of sensitivity values are statistically insignificant (Supplementary Table S8).

Figure 5 and Supplementary Table S7 also highlight two observations. First, GTalign's performance diminishes when employing sequence prefiltering (options `--speed=13 --pre-similarity=15`), due to overly stringent criteria in filtering out dissimilar structures, despite relatively high alignment accuracy for pairs passing the filter (Figs. 1, 2, and 4). Future enhancements may involve refining sequence prefiltering options.

Second, disregarding cross-fold relationships considerably affects precision, recall, and AUPRCs, rendering evaluations unstable. Further discussion on this aspect follows in the subsequent subsection.

## Limitations of SCOPe-based evaluation

In the SCOPe classification, the traditional emphasis on organizing proteins according to their sequence similarity to classified entries[36] reflects a primary focus on evolutionary relationships. However, structural similarities among proteins extend beyond evolutionary connections[37]. Classifying proteins into discrete

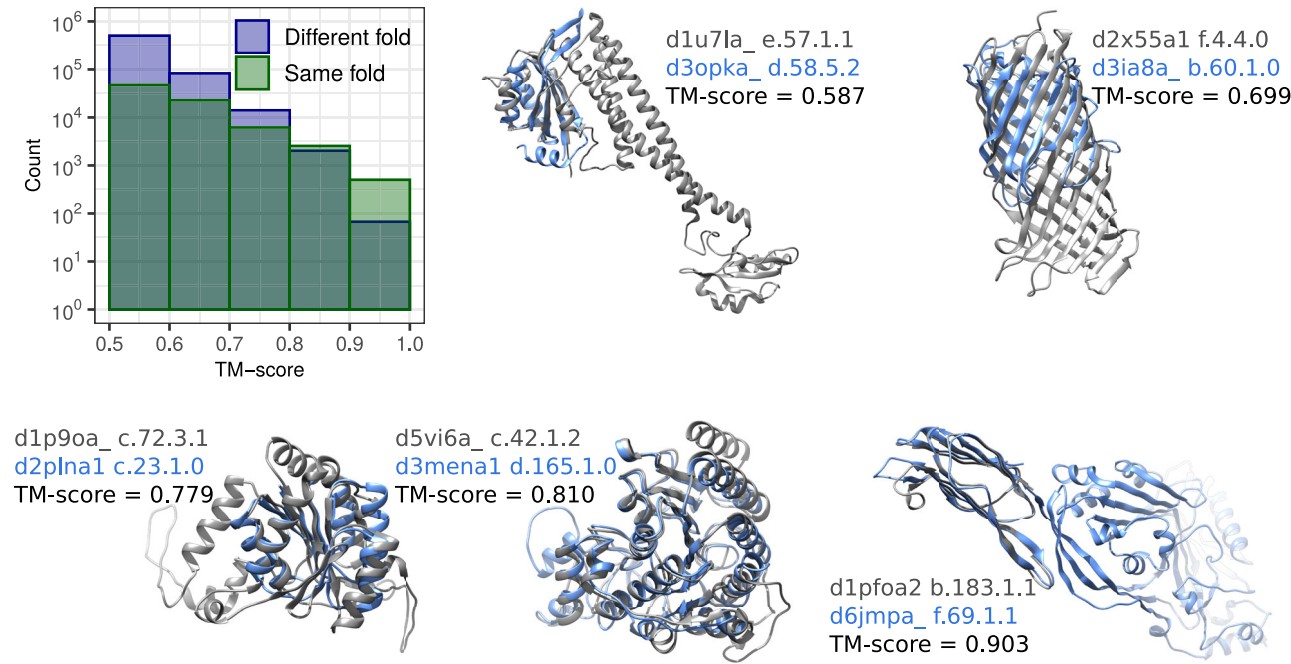

**Fig. 6 | Distributions of TM-scores (>0.5) for domain pairs within the same and different SCOPe 2.08 folds, presented in bins of width 0.1.** TM-scores, normalized by the length of the shorter protein, were calculated by aligning query and subject structures from the SCOPe40 2.08 dataset using TM-align. Representative examples of structure pairs from different folds and classes are provided for each TM-score bin. These examples, along with numerous other significantly structurally similar pairs from different folds and classes, are considered errors (false positives) in the reference SCOPe-based evaluation. Source data are provided as a Source data file.

folds presents challenges due to inherent ambiguity in defining folds[38,39]. For example, what level of insertions or deletions can be considered critical for classifying a domain into a different fold? A more nuanced perspective suggests that protein fold space may exhibit both discrete and continuous characteristics[37,39]. While high structural or evolutionary similarities may support discrete fold assignments, lower yet significant similarities imply a continuous nature of fold space.

Our analysis supports these assertions. Examination of the distribution of statistically significant TM-scores, obtained by aligning query and subject structures from the SCOPe40 2.08 dataset using TM-align, reveals that only highly similar protein domains (TM-score >0.8) within the same folds outnumber those from different folds (Fig. 6, Supplementary Fig. S8). Domains from different folds span the entire TM-score significance range, with all of them deemed errors (false positives) despite evident structural and topological similarities (Fig. 6).

These insights underscore the limitations of the SCOPe-based evaluation. While the reference SCOPe-based evaluation provides a convenient approach for benchmarking structure alignment tools, its inability to capture the full complexity of protein structure space is evident.

In conclusion, GTalign provides an efficient solution for searching vast protein structure datasets at different levels of accuracy. Its high efficiency is exemplified by a speedup of 6 orders of magnitude over TM-align when aligning large protein complexes (Supplementary Fig. S9). GTalign's cross-platform implementation, user-friendly interface, and high configurability, including the option for clustering structures (Supplementary Section S1.6), underscore its accessibility and versatility. Providing orders of magnitude in speedup at state-of-the-art accuracy, GTalign positions itself as a valuable tool among existing structure aligners.

## Methods

### Structure representation
GTalign offers users the flexibility to configure and choose which protein structure atoms will serve as representatives. By default, protein structures are represented using alpha-carbon atoms. All experiments conducted with GTalign were performed using alpha-carbon atoms as representatives.

### Algorithm outline
The GTalign software takes as inputs query and subject (referred to as "reference" in the software) structure databases of arbitrary size. GTalign processes this data in chunks, aligning batches of query structures with batches of subject structures, both sorted by length, iteratively until all possible batch pairs are completed. A similar batch-oriented approach to processing large databases has been described previously[27]. Below, we outline the (sequential) algorithmic steps representing the actions performed on a pair of batch query and subject structures.

Certain steps in the algorithm involve the alignment refinement procedure described in Algorithm 10 (RefineBestAlignments) specified in Supplementary Section S2. Algorithm 10 optimizes TM-scores and refines alignments by considering differently positioned alignment fragments of different lengths. It takes three parameters: the numbers of query and subject structures, where all possible pairs are processed in parallel, and a gap opening penalty for the COMER2 DP algorithm[27] to generate alignments that optimize TM-scores given the superpositions. The complete outline of the GTalign algorithm is provided below.

1. Index query and subject structures and store the spatial indices in a $k$-d tree data structure.
2. Assign secondary structure states to the structures at each residue in parallel. This assignment is determined by the coordinates of five residues centered around the residue under consideration, with distance cutoffs between residues optimized in ref. 21.

3.  Calculate transformation matrices based on continuous fragment pairs in parallel for all queries and subjects, their matched fragment pairs, and fragment positions.

4.  Apply Algorithm 10 with parameters $(n_Q, n_S, 0)$ to refine alignments obtained from the superpositions found in the previous step to maximize the TM-score (always normalized by the length of the shorter protein). $n_Q$ and $n_S$ are the numbers of query and subject structures in batches. Here and in the following steps, keep track of the maximum TM-score and the corresponding transformation matrix for all query-subject pairs.

5.  If the option `--add-search-by-ss` is specified, apply Algorithm 10 with parameters $(n_Q, n_S, \{-0.6, 0\})$ to refine alignments obtained from the application of the COMER2 DP algorithm[27] using a scoring function based on secondary structure matching and sequence similarity score[40].

6.  Apply Algorithm 1 (DeepSuperpositionSearch) to find the most favorable superpositions through a deep search using spatial indices. The search depth is controlled with the `--speed` option. This step is central to the GTalign method because it enables rapid exploration of the superposition space, resulting in accurate alignments. We provide a detailed specification of Algorithm 1 in Supplementary Section S2.

7.  Apply Algorithm 10 with parameters $(n_Q, n_S, \{-0.6, 0\})$ to refine alignments obtained from the application of the COMER2 DP algorithm using a scoring function based on secondary structure matching and TM-score, given the optimal transformation matrices obtained so far.

8.  Apply Algorithm 10 with parameters $(n_Q, n_S, \{-0.6, 0\})$ to refine alignments obtained from the application of the COMER2 DP algorithm using TM-score as a scoring function. Here, the number of repetitions in Algorithm 10 is configurable (option `--convergence`).

9.  Produce final alignments using the COMER2 DP algorithm based on the optimal transformation matrices in parallel for all queries and subjects.

10. Calculate TM-scores, root-mean-squared differences (RMSDs), and other alignment-related statistics in parallel for all queries and subjects.

Steps 1 and 2 prepare data for processing. Steps 3 and 4 identify protein superpositions by matching continuous protein segments, similar to TM-align's initial gapless matching[21]. These steps are sufficient to capture optimal superpositions of proteins sharing high structural similarity over a significant fraction of the length of at least one protein of a pair. Step 5 occasionally improves superpositions found in the previous steps. Step 6 conducts an extensive superposition search by matching different protein spatial regions. Steps 7 and 8 represent the refinement of transformation matrices and related alignments obtained earlier, meaning that alignment regions typically do not change or change slightly. Steps 9 and 10 prepare results for output. All the steps are based on algorithms and data structures designed to maximize instruction and memory throughput.

### Spatial index data structure
To accelerate the superposition search, protein structures are initially indexed (step 1 of the algorithm outline). Each structure's index is stored in a $k$-d tree data structure, which hierarchically organizes protein atom coordinates. This organization allows for the retrieval of the nearest neighbor in the tree for a query atom with specified coordinates in constant, $O(1)$ time.

### Accelerated superposition search using spatial indexing
The accelerated superposition search process (step 6 of the algorithm outline) leverages spatial indexing to find optimal superpositions for query and subject proteins within a data chunk. Conducted in parallel for all query-subject protein pairs in the chunk, this process explores numerous initial superposition configurations per protein pair simultaneously.

Initially, the process calculates initial superpositions, or transformation matrices, based on continuous query and subject protein fragments spanning the entire extent of both proteins. The search depth, determining the number of superpositions to explore, and the fragment length depend on the query and subject protein lengths, with the fragment length not exceeding 100 residues. Regions with low local secondary structure similarity between query-subject fragment pairs avoid the calculation and exploration of initial superpositions.

Upon completing initial superpositions, the shorter protein undergoes transformation to obtain spatial overlays. Then, alignments are generated between the shorter protein and the other using the longer protein's index in parallel over residues, achieved in constant time complexity. This routine repeats twice: Initially produced alignments refine spatial overlays, followed by repeated alignment production using the protein index while ensuring matching protein secondary structure this time.

The most favorable alignment for the query-subject pair is then selected based on the highest TM-score. However, alignments obtained using spatial indices are sequence order-independent. Therefore, approximate sequence order-dependent TM-scores are computed from these alignments, with one structure transformed, in sub-linear time considering a maximum of 512 aligned residues.

Next, a small subset of transformation matrices with the highest approximate scores is chosen for TM-score calculation using the COMER2 DP algorithm. Further refinement involves selecting an even smaller subset of transformation matrices corresponding to the highest TM-scores to optimize alignments, considering different-length and differently positioned alignment fragments. Finally, the best alignment for the query-subject pair is selected and refined similarly, employing full DP and TM-score optimization.

A detailed specification of this procedure is provided in Supplementary Section S2.

### Calculation of rotation matrices
GTalign computes rotation matrices using the Kabsch algorithm[41,42]. Solving for the eigenvalues and eigenvectors of a cross-covariance matrix, $K \in \mathbb{R}^{3 \times 3}$ ($R$ in the original notation), requires double-precision arithmetic. To render the problem solvable in single precision, thereby boosting instruction and memory throughput, $K$ is normalized by the mean of the absolute values of its elements. It is easy to show that this operation corresponds to scaling the coordinates of protein atoms. Effectively, rotation matrices for large proteins can be considered as obtained using coordinates expressed in nanometers instead of Angstroms, preventing single-precision arithmetic overflow and underflow. The resulting rotation matrices exhibit an insignificant error (on the order of $10^{-5}$ on average) with no discernible impact on superposition and structural alignment while still ensuring high performance.

### Dynamic programming implementation
A previously published algorithm[27] was employed to implement the dynamic programming (DP) algorithms. The time complexity to calculate DP matrices is $O(\max_q l_q + \max_s \tilde{l}_s)$, with a constant factor dependent on the number of threads running in parallel. (The computation involves $(\max_q l_q + \max_s \tilde{l}_s)/32$ iterations of independent and parallelized calculations, executed in $O(32)$ time by GPU threads.) Here, the maximums are taken over the lengths of all query ($l_q$) and subject ($\tilde{l}_s$) proteins in a data chunk. In instances where DP matrix values are only required to update backtracking information, the memory complexity for DP matrices is $O(n_Q \sum_s \tilde{l}_s)$[27], with $n_Q$ representing the number of query proteins in the chunk.

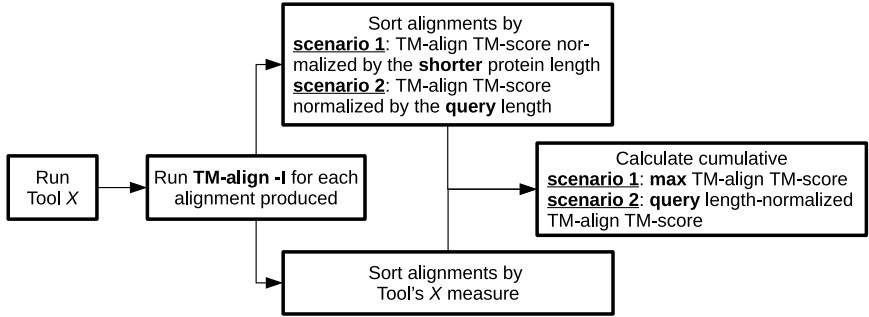

**Fig. 7 | Schematic for benchmarking structure alignment tools.** The entire procedure can be described as follows: (i) Run a structure alignment tool; (ii) Use TM-align to calculate the TM-score for each produced alignment; (iii) Sort the alignments by the tool's measure (e.g., *P*-value, *Z*-score, etc.); (iv) In addition, sort the alignments by the TM-score calculated by TM-align; (v) Finally, calculate the cumulative TM-score for the results, considering both the sorting by the tool's measure and the sorting by the TM-align-obtained TM-score. This provides a comprehensive measure of how accurately the tool produces alignments and their rate. It's worth noting that seemingly subtle differences in cumulative TM-score can be significant, especially considering the narrow gap between successive TM-scores; a mere 0.2 difference can distinguish between an accurate and inaccurate alignment.

DP matrices are built using a gap opening penalty, while the gap extension cost is set to 0 unless otherwise specified. For optimizing memory usage, match scores are modified before writing to memory— either by negating them for non-negative alignment scores or subtracting a large constant for potentially negative scores. The reverse operation is subsequently applied upon reading from memory. This approach minimizes memory requirements, facilitating greater data accommodation and parallelization.

### Algorithm efficiency

The high efficiency of the developed algorithms, particularly the spatial indices, is best demonstrated by aligning large protein structures. For instance, GTalign only took seconds to align and provide a superposition for two virus nucleocapsid variants, 7a4i and 7a4j (37,860 residues each), featuring different chain orders on a single Tesla V100 GPU (Supplementary Fig. S9). In contrast, aligning these complexes using TM-align took more than three months. Although TM-align is not typically used for aligning complexes, GTalign's efficiency may open additional possibilities for exploring large complexes when chain order preservation is important.

### Performance improvement potential

GTalign's efficiency can be further enhanced by considering three key aspects. First, GTalign currently uses 32-bit floating-point precision (FP32) operations. Exploring the adoption of 16-bit (FP16) or even 8-bit (FP8) floating point precision before the final stages of alignment has the potential to increase the degree of parallelization by 2 to 4-fold.

Second, the COMER2 DP algorithm, a critical component for accuracy, is employed several times throughout the structural alignment search procedure. Substituting it with spatial matching, as outlined in Supplementary Section S2, at all intermediate stages and reserving it solely for the final alignment stage could result in a significant speedup.

Finally, the third aspect involves similarity selection on the coarse scale. By encoding structures with embeddings and utilizing indexed vector databases[43], GTalign could achieve nearly instantaneous selection of similar protein candidates and a constant-time database search and alignment, regardless of the database size.

### Prescreening for similarities in sequence and structure space

GTalign allows for an initial screening in the sequence space (option `--pre-similarity`) to identify potential similarities before engaging in more detailed structural analysis. The implementation of this procedure is based on calculating local ungapped alignment scores between protein sequences using a sequence similarity score table[40] and does not involve dynamic programming. Protein pairs with alignment scores exceeding a specified threshold progress to the subsequent stages of structural analysis.

In addition, an initial screening for similarities is available in the structure space using the `--pre-score` option. With this option, protein pairs with provisional TM-scores, obtained in step 4 of the algorithm outline, lower than a specified threshold are excluded from further processing.

### GTalign software

GTalign incorporates several key features that contribute to its versatility and user-friendly nature. Developed using the OpenMP standard for CPUs and CUDA architecture for GPUs, GTalign is compatible with various computing architectures, including NVIDIA Pascal, Turing, Volta, Ampere, Ada Lovelace, and subsequent GPU architectures. (The GPU version exhibits a 10–20x increase in speed.) Its independence from external packages ensures seamless operation across different compilers (GCC, LLVM/Clang, MSVC) and their respective versions. GTalign is cross-platform software, with binary packages precompiled for Linux and Windows x64 operating systems. For other platforms, users have the flexibility to compile GTalign from its source code. GTalign usage is straightforward: No structure database preprocessing is required. Users can effortlessly employ GTalign by directly providing files, compressed files (`gzip`), directories, and/or archives (`tar`) of protein structures as command-line arguments. This user-centric design enhances accessibility and facilitates streamlined integration into diverse computational environments.

### Alignment accuracy evaluation

The evaluation of structural alignment accuracy is based on assessing how accurately the structural alignments of protein pairs translate to spatial agreement in their respective structures. This self-contained evaluation is unbiased, as it does not depend on external classifications, which may be constructed using specific sequence and structure alignment tools.

The superposition of two aligned proteins is evaluated by the TM-score and RMSD, calculated by the established method TM-align[21] using the `-I` option (Fig. 7). Notably, in this setting, TM-align does not perform a global superposition search but instead optimizes superposition constrained by a given alignment, leaving it unchanged.

GDT_TS scores were calculated using the TM-score tool[26], with minimal modifications to the source code to normalize GDT_TS by the number of aligned residue pairs. The adapted TM-score code is publicly available.

In the benchmarks, alignments are evaluated based on (i) the TM-score normalized by the length of the shorter protein and (ii) the TM-score normalized by the query length. The first scenario considers all structural similarities, including instances where smaller proteins match regions of larger proteins. The second scenario downgrades the importance of alignments between the query and a much shorter subject protein, providing a more favorable position for some methods (e.g., Dali[22]) as their measures (e.g., Z-score) reduce the significance of such alignments.

## SCOPe-based evaluation

This evaluation aimed to assess the ability of the tools to match SCOPe 2.08[29] domains to families, superfamilies, and folds. True positives (TPs) at the family, superfamily, and fold level were defined as pairs of structures from the same family, the same superfamily but different families, and the same fold but different superfamilies, respectively. Self-matches were excluded. The sizes of these groups are referred to as effective sizes. False positives (FPs) were identified as pairs from different folds.

Precision and recall were calculated as #TP/(#TP + #FP) and #TP/P, respectively, where P represents the total number of positive pairs. The number of TPs, #TP, and P for precision-recall (PR) curves were downweighted by the effective size of family, superfamily, and fold for respective-level calculations. The number of FPs, #FP, was downweighted by the effective fold size. The weighting for counts was consistent with the approach used in ref. 25.

Before conducting sensitivity and PR analyses, alignments generated by the tools were sorted by their significance measure. Foldseek (default parametrization) alignments were sorted by E-value, while FATCAT alignments were sorted by P-value, and Dali alignments by Z-score. DeepAlign alignments were sorted by DeepScore. TM-align and GTalign alignments were sorted by the harmonic mean of the TM-scores normalized by the query and subject lengths. The harmonic mean proved superior to the arithmetic mean for TM-align and GTalign alignments due to its ability to reduce significance for structure pairs with large length differences. However, the arithmetic mean was more suitable for Foldseek `--tmalign-fast 1` and `--tmalign-fast 0` alignments, as most of such pairs had already been filtered out.

Secondary TM-scores, referred to as 2TM-scores, were introduced to rank GTalign alignments in the SCOPe-based evaluation. The 2TM-score is calculated over the alignment excluding unmatched helices and provided slightly improved results for fold-level evaluations. Options to calculate 2TM-scores (`--2tm-score`) and rank alignments by the harmonic mean of the TM-scores or 2TM-scores are available starting with version 0.15.0.

## The SCOPe40 2.08 dataset

All protein domains from the SCOPe 2.08[29] database filtered to 40% sequence identity (SCOPe40 2.08), totaling 15,177, were searched with query protein domains selected randomly, one per superfamily, from the same SCOPe40 2.08 dataset. Representatives that Dali[22] failed to reformat for its initial structural representation were omitted, resulting in a total of 2045 queries.

To ensure consistent structure interpretation between TM-align and the other tools, the structure files underwent the following changes: (i) the first model of multi-model files was retained; (ii) the chain identifier was set to 'A' to make a single-chain structure; (iii) residues were renumbered sequentially; (iv) residues lacking at least one of the N, CA, C, and O atoms were removed. HETATM records were disregarded when using Foldseek as its interpretation of these records differed from that of TM-align.

## The PDB20 dataset

PDB[30] structures filtered using `blastclust`[44] version 2.2.26 to 20% sequence identity with a length coverage threshold of 70% (PDB20),

totaling 18,801, were queried with 186 CAMEO protein structure targets[45] released over 3 months from 07/24/2021 through 10/16/2021. The CAMEO targets and the PDB20 structures maintained no more than 20% sequence identity.

The structure files were preprocessed to ensure consistent structure interpretation across the tools: HETATM records and residues lacking at least one of the N, CA, C, and O atoms were removed. Also, the first model of multi-model files was retained.

## The Swiss-Prot dataset

All UniProtKB/Swiss-Prot[31] protein structures (542,378) from the AlphaFold Database[4] were searched with 40 proteins representative of structurally diverse CRISPR-Cas systems[46]. The selection of the 40 query proteins followed a specific process: First, the 5831 PDB protein chains associated with CRISPR-Cas systems (downloaded on 10/19/2023) were clustered at a TM-score threshold of 0.4 with a length coverage threshold of 40% using GTalign with options `--speed=13 --add-search-by-ss --cls-coverage=0.4 --cls-threshold=0.4 --ter=0 --split=2`. Subsequently, the top 40 members from every third singleton cluster, sorted by length, were chosen as queries, with an average length of 382 residues. The query structures underwent preprocessing, involving the removal of HETATM records and residues lacking at least one of the N, CA, C, and O atoms.

## The HOMSTRAD dataset

The HOMSTRAD dataset, comprising reference structural alignments of protein families and accompanying structure files, was obtained from ref. 34, containing 398 multiple protein structure alignments from the HOMSTRAD database[32]. (The original data were inaccessible). For benchmarking purposes, each family's first protein from the reference alignments was aligned with every other protein of the same family, resulting in a total of 1722 pairwise alignments.

## Computer system configuration

Unless otherwise specified, all benchmark tests were conducted on a server equipped with two Intel Xeon Gold 5115 CPUs @ 2.4 GHz (20 hardware threads per CPU), 128GB DDR4 RAM, and three NVIDIA Tesla V100-PCIE-16GB GPU accelerators, running the CentOS 7 operating system.

## Runtime evaluation

The runtimes of all tools were measured by the Linux `time` command.

## GTalign settings

Unless otherwise specified, all analyses were performed using GTalign version 0.14.0, compiled with GPU support. For protein pairs in the HOMSTRAD dataset, alignments were generated with the command

```
gtalign --qrs=<query_file> --rfs=<subject_file> -o <output_dir> --hetatm --dev-min-length=3 --speed=<speed> --pre-score=0 -s 0 --add-search-by-ss --dev-mem=4096
```

The command used to process queries for the SCOPe40 2.08, PDB20, and Swiss-Prot datasets was

```
gtalign --qrs=<query_dir> --rfs=<db_dir> -o <output_dir> --hetatm --dev-queries-total-length-per-chunk=1500 --dev-min-length=3 --dev-max-length=<max_len> --speed=<speed> -s 0.44 --add-search-by-ss --nhits=<n_hits> --nalns=<n_hits> --dev-N=3
```
where `<query_dir>` represents the directory of query structures, `<db_dir>` denotes the directory or `tar` archive (Swiss-Prot dataset) containing subject structures, `<max_len>` is 5000 for the PDB20 dataset and 4000 for the others, and `<n_hits>` is 10,000, 4000, and 50,000 for the SCOPe40 2.08, PDB20, and Swiss-Prot datasets, respectively. The value of the `-speed` option, along with the usage of several additional

options (`--pre-score` and `--pre-similarity`), is indicated in Figs. 1–5 and Supplementary Tables and Figures. When employing the initial screening in the sequence space (option `--pre-similarity`), options `-s 0.3` and `-c cachedir` were specified. The latter was used unconditionally for the Swiss-Prot dataset. In the SCOPe-based evaluation, the `--2tm-score` option was specified to calculate 2TM-scores (version 0.15.0).

GTalign calculates and outputs TM-scores normalized by the length of both proteins in a pair. Consequently, the corresponding TM-scores were utilized as GTalign's measures to sort alignments (left panels of Figs. 1b–d and 2a–c).

## TM-align settings

Parallel processing of all queries for each dataset was achieved by iteratively running 40 instances of TM-align[21] version 20220412 simultaneously. For the HOMSTRAD and three other datasets, each process instance was executed with the following options, respectively: `<query> <subject_file> -het 1` and `<query> -dir2 <db_dir> <lst_file> -het 1`. Here, `<query>` represents a query file, `<db_dir>` is a directory of subject structure files, and `<lst_file>` is a list file of all subjects. For the Swiss-Prot dataset, `-outfmt 2` was included to reduce disk space usage. The fast version (TM-align -fast) utilized an additional option, `-fast`.

## Dali settings

The standalone version DaliLite.v5[22] was employed in the benchmark tests. Prior to initiating searches, structure files underwent reformatting to an initial representation using the command `import.pl --pdbfile <struct_file> --dat <dir> --pdbid <id>`. In this command, `<struct_file>` refers to a structure file, `<dir>` is a directory for reformatted structures, and `<id>` is an assigned structure identifier. Reformatting failed for 104 and 525 subject structures from the SCOPe40 2.08 and PDB20 datasets, respectively. The time taken for reformatting was excluded from runtime evaluations.

For parallelizing searches, 40 process instances were executed using the command `dali.pl --np 40 --query <query_list> --db <sbjct_list> --dat1 <query_dir> --dat2 <db_dir> --outfmt alignments`. In this command, `<query_list>` and `<sbjct_list>` represent the list files of query and subject structures, respectively, while `<query_dir>` and `<db_dir>` indicate the directories of (reformatted) query and subject structures. Dali's Z-score served as the sorting measure, arranging the alignments in descending order based on it.

For the HOMSTRAD dataset, reformatting was not used, and alignments were directly generated with the command `dali.pl --pdbfile1 <query_file> --pdbfile2 <subject_file> --dat1 <query_dir> --dat2 <subject_dir> --outfmt alignments`.

## DeepAlign settings

All query-subject structure pairs in each dataset underwent parallel processing through the iterative execution of 40 simultaneous instances of DeepAlign[18] version v1.4 Aug-20-2018 (https://github.com/realbigws/DeepAlign), using the command `DeepAlign <query_file> <subject_file>`.

DeepAlign outputs the TM-score normalized by the length of the shorter protein, which was used as its measure for sorting alignments in the corresponding evaluations (left panel of Fig. 1b, c). When evaluating alignments based on the TM-score normalized by the query length, DeepAlign's DeepScore was utilized as its measure to sort the alignments (left panel of Fig. 2a, b).

## FATCAT settings

FATCAT 2.0[28] searches were conducted iteratively for all queries in three datasets using the rigid structural alignment setting. The command `FATCATSearch <query_file> <sbjct_list> -i2 <db_dir>` `-r -o <output_file> -m` was utilized, with `<sbjct_list>` and `<db_dir>` representing the list file and the directory of subject structures, respectively. For each query, FATCAT automatically initiated parallel processes corresponding to the number of processors in the system, in this case, 40. FATCAT's P-value served as the sorting measure, i.e., the alignments were sorted in ascending order based on it. It is noteworthy that FATCAT disregards HETATM records, and thus, these records were also omitted during alignment accuracy evaluation with TM-align.

For the HOMSTRAD dataset, alignments were generated with the following command: `FATCAT -p1 <query_file> -p2 <subject_file> -r -o <output_file> -m`.

## Foldseek settings

Foldseek[25] version `d1d1b868a571a9a0c62ae50b07139ebdd224f879` was used (downloaded on 6/25/2023). All queries in the SCOPe40 2.08, PDB20, and Swiss-Prot datasets were parallelized using the following command:

```
foldseek easy-search <query_dir> <db_dir> <output_file>
<tmp_dir> --threads 40 --max-seqs 4000 --format-output
query,target,evalue,qtmscore,ttmscore,alntmscore,
rmsd,qstart,qend,qlen,tstart,tend,tlen,qaln,taln
```

In this command, `<query_dir>` denotes the directory of query structures, `<db_dir>` indicates the directory or `tar` archive (Swiss-Prot dataset) containing subject structures, and `<tmp_dir>` is a temporary directory. For the Swiss-Prot dataset, the option value `-max-seqs 20000` was specified. Foldseek was additionally parameterized with the options `--alignment-type 1` and `--tmalign-fast 0` to enable the use of the fast (Foldseek --tmalign-fast 1) and regular (Foldseek --tmalign-fast 0) versions of TM-align for alignment production.

As Foldseek generates TM-scores, the TM-score normalized by the length of the shorter protein was employed as the sorting measure for alignments in the left panel of Fig. 1b–d. When evaluating alignments based on the TM-score normalized by the query length (Fig. 2a–c), E-value and the average TM-score (Foldseek --tmalign-fast 0/1), a recommended metric[25], were utilized as measures to sort the alignments in ascending and descending order, respectively.

For the HOMSTRAD dataset, the command was modified to specify individual query and subject structure files instead of their directories. Also, the option `-threads` was set to 1, and the additional options `--prefilter-mode 2` and `-e 1e6` were included.

## Figure preparation

Molecular graphics images were generated using UCSF Chimera[47] version 1.14. Plots were created using the ggplot2 package[48] in R[49], versions 3.6.0 and 4.3.2.

## Reporting summary

Further information on research design is available in the Nature Portfolio Reporting Summary linked to this article.

## Data availability

The SCOPe 2.08 PDB-style files with coordinates for the SCOPe40 2.08 dataset are available at https://scop.berkeley.edu/downloads/. The PDB files for the PDB20 dataset are available at https://www.rcsb.org/downloads. The archive of the Swiss-Prot protein structures is available at https://ftp.ebi.ac.uk/pub/databases/alphafold/latest/swissprot_pdb_v4.tar. The HOMSTRAD dataset, originally obtained from http://yanglab.nankai.edu.cn/mTM-align/benchmark, is available at https://github.com/minmarg/gtalign-evaluation. The benchmark data generated in this study have been deposited in the Zenodo

database (https://doi.org/10.5281/zenodo.11148017). Source data are provided with this paper.

## Code availability

The source code and software packages of GTalign are available at https://github.com/minmarg/gtalign_alpha, and the corresponding releases are archived on Zenodo: https://zenodo.org/records/10433420 (version 0.14.0; https://doi.org/10.5281/zenodo.10433420) and https://zenodo.org/records/10433419 (versions >0.14.0; https://doi.org/10.5281/zenodo.10433419)[50]. The lists of structure identifiers, as well as scripts and programs for structure preprocessing, benchmarking, and producing graphs, are available at https://github.com/minmarg/gtalign-evaluation.

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

## Acknowledgements

This work was inspired by the success of TM-align[21]. The research was supported by Research Council of Lithuania (LMTLT) grant S-MIP-23-104 (M.M.).

## Author contributions

M.M. conceived and supervised the project, designed the algorithms, developed GTalign, conducted the analyses, and wrote the manuscript.

## Competing interests

M.M. is a co-founder of Hiomics digital and declares non-financial competing interests.
