## [Peer Review File · Nature Communications]

GTalign: Spatial index-driven protein structure alignment, superposition, and searchREVIEWER COMMENTS

Reviewer #1 (Remarks to the Author):

This is an outstanding contribution that demonstrates truly impressive speed up of structural alignments based on the TM-score.

The work is exceptionally well documented and carefully done and merits publication without revision. This will prove to be a very useful tool to the structural biology community.

Reviewer #1 (Remarks on code availability):

Code is correct and very easy to follow.

Reviewer #2 (Remarks to the Author):

The manuscript submitted by Margelevicius entitled "GTalign: High-performance protein structure alignment, superposition, and search" describes a new algorithm for protein structure alignment, which, based on presented results, is on par with state-of-the-art structure alignment methods like TM-align in terms of accuracy while performing at much higher speed. While GTalign is an order of magnitude slower than Foldseek, based on the results presented, the accuracy of Foldseek is considerably lower. The presented tradeoff of GTalign in terms of accuracy and speed represents a major improvement in the field and thus, the presented results are expected to be of high interest to the field of computational structural biology.

Major comments:

While the manuscript is in principle well written in terms of English, it is very hard to understand conceptually the algorithmic approach and how this new algorithm is different compared to previously published approaches. At various places in the manuscript (abstract, introduction, results, and discussion) clarity of the manuscript could be improved by providing a proper introduction of the current state-of-the-art, conceptual layout of the novel approach, and discussion of the results with respect to previously published findings. The author should keep in mind that readers of Nature Communications come from a wider field of research but in its current state, the manuscript can only be understood by experts in dynamic programming, k-d tree data structure searches, etc.

While the methods section, i.e. 4.2, is an attempt to more conceptually introduce the approach, it is not understandable because it lacks information on one side (i.e. "Assign secondary structure states to the structures based on the parameters optimized in [12]." -> so readers would need to now read upon reference 12 to understand what is done here?) while on the other hand suddenly provides a lot of technicalities that are not explained or are explained in the supplementary text (i.e. "Apply Algorithm 10..."). Furthermore, it is also not clear if all 10 steps presented in 4.2. are executed one after the other or represent alternative executions.

Fig 1a is supposed to illustrate the approach but lacks any conceptual detail and thus is not helpful to support understanding by the reader.

The moderate performance of Foldseek in the various benchmark datasets is surprising, especially as it seems that essentially identical benchmark datasets were used here and in Kempen et al Nature Biotechnology 2023. In Kempen et al, Foldseek performed on par with methods like TM-align. It is clear that different approaches were used to evaluate performance of the various methods in Kempen et al and here, however, to improve clarity and comparability of the results presented here, it would be advisable to also perform evaluation of performances similar to approaches presented in Kempen et al, for example. If the author thinks that the evaluation procedure presented here is more accurate or relevant compared to evaluation procedures used elsewhere, then this should be laid out and

supported with data.

Minor comments:

References to figure panels are not in order (i.e. Fig 2 is referenced prior to Fig 1b) or missing (i.e. Fig S1).

The different colors in panels of Fig 1 are very hard to distinguish. This is only possible upon close zoom in. Visualization could be improved either by labelling the different lines in the plot or by adding zoom-ins, for example.

It would be helpful to add to the legend of Fig 1 why different x and y scales were used in panels b-d. I assume this is because of the different dataset sizes but it would be good to clarify in the legend how the axis ranges were set.

Fig 1d. Why does Foldseek not extend to the right end of the x-axis range while the other tools do?

Page 5 lines 183-187 are unclear. What does the author mean by "spectral analysis 184 of geometrical features in the frequency domain (as explored in our in-house research)"?

It is not clear what the author refers to when talking about the sequence prefiltering step for Foldseek. Also, can the author please better explain the two Foldseek options talign-fast 0 and 1?

Reviewer #3 (Remarks to the Author):

Margelevicius et al. describe GTalign, a new method to compute pairwise protein structure alignments. The algorithm describes an iterative alignment algorithm utilizing spatial indices to improve search speed. The software is optimized for speed by comparing structures in batches and by utilizing GPU processing capabilities. A standout feature is how smoothly the software install went and that everything worked as expected with clear documentation.

The algorithmic idea of using spatial indices and the capability of batching multiple queries at once seem interesting and novel, but it would be great if they were discussed more in detail in the main manuscript in a broadly accessible way.

However, my main critique centers on the benchmarking of the software. First, regarding performance benchmarks: GTalign does not compare as favorably to TAlign as stated in the manuscript during my evaluation of the software on an A5000 GPU. Next, regarding sensitivity: The shown reference-free sensitivity benchmark results do not match the reference-based benchmark. This could be due to the reliance on a hard TMscore cut-off of 0.5, which is not a good discriminator whether a match is a true positive or false positive (at least according to community standards such as SCOP/CATH). The examples chosen to highlight the strength of the method in Supp. Fig. 2 are excellent examples why solely relying on a TMscore of >0.5 is insufficient to detect similar folds reliably, since all shown example alignments are from different folds, with two even belonging to different architectures according to the SCOP hierarchy. Additionally, I have concerns regarding circularity of both conducting iterative TMscore optimization and using TMscore as an evaluation criterion.

Major:

- Sensitivity benchmark:

The benchmark has two major issues: The chosen measure of cumulative TMscore can become highly inflated by (over-)counting false positive alignments, which results in non-meaningful reported matches and harms the performance of methods that carefully reject false positive matches.

Additionally, a hard acceptance threshold of >0.5 TM-score allows alignments of proteins that do not share evolutionary history according to SCOP or CATH. SCOP and CATH classification on family,

superfamily, and fold level are community standards to define evolutionary relationships and are commonly used to benchmark aligners. While the SCOP benchmarks shown in the manuscript use proteins contained within SCOP, they do not leverage the SCOP hierarchy.

Below is a table of average ROC1 (Fraction of TPs up to the first FP) values for family, superfamily, and fold following the SCOP benchmark proposed by Foldseek (van Kempen et al. 2023). From these numbers it can be seen that DALI performs best, followed by TAlign, GTalign, Foldseek-TM in a similar range, and Foldseek in last position. This stands in contrast with the results presented. Additionally, the table is annotated with the runtime of each method (see below).

Methods	Family	S.fam.	Fold	Speed in seconds
DALI	0.888076	0.575125	0.165703	123010
TAlign(fast)	0.892301	0.55768	0.163492	62240
Foldseek-TM	0.881055	0.547637	0.163978	1889
GTalign	0.88417	0.54846	0.157763	59748
Foldseek	0.86622	0.492773	0.108328	42

- The given example structures in Supplemental Fig. S2 should not be well alignable according to SCOP. Two out of the five examples are from different SCOP architectures (d2vv5a2, d4i43b4; d7coha_, d2hkta_) while the other three are in different folds. When looking at the examples d7coha_ and d2hkta_ it is visually clear that they have different helix architectures and should not be aligned with a high similarity score. All other aligners seem to agree that these structures shouldn't be aligned.

d1aoaa1 a.40.1.1 d1c3ca_ a.127.1.1 Different Fold
d1e8ca1 c.98.1.1 d1enoa_ c.2.1.2 Different Fold
d1x3la_ c.118.1.0 d2p9ja_ c.108.1.0 Different Fold
d2vv5a2 d.58.43.1 d4i43b4 e.8.1.8 Different Architecture
d7coha_ f.24.1.1 d2hkta_ a.285.1.1 Different Architecture

- TMscore circularity:

Conducting iterative TMscore optimization and evaluating with TMscore as a cut-off in benchmarks seems circular and unfair. Choosing an independent measure (e.g. DALI Z-score) from the one being optimized should make the comparison much fairer.

- Speed:

When benchmarking GTalign on an Nvidia A5000 GPU, I did not see speed advantages over TAlign, compared to paper's claim of a one order of magnitude speedup over TAlign-fast. Here, TAlign (fast) was benchmarked on a 64-core machine. Foldseek seems to be also four orders of magnitude faster compared to GTalign (default) and not just one as claimed. How can this discrepancy be explained?

I executed GTalign with the following command:

```
gtalign -v -qrs=scop40.tar --rfs=scop40.tar -o test -s 0.
```

- Appropriate hardware configuration for benchmarking:

GTalign was benchmarked on three V100 GPUs and is compared against CPUs with 40 hardware threads in total. While it is difficult to fairly benchmark different hardware architectures in general, this evaluation does not seem fair from a performance/\$ perspective.

The approximate cost of the two CPUs used should be 2000-3000\$ depending on the acquisition date, while the three V100 GPUs cost approximately 6000-30000\$ depending on when they were acquired. The fairest benchmark, given these prices, while keeping the same hardware configuration, would be to compare the two CPUs with 40 hardware threads vs. one V100 GPU.

- Explanation of algorithmic novelty

In its current form, the algorithmic novelty may not be easily comprehensible. Although the authors have included pseudocode in the supplementary materials, it remains challenging to understand. I suggest explaining, in particular, the constant-time comparison in detail, as this appears to be one of the most, if not the most important innovations of the work.

Reviewer #3 (Remarks on code availability):

I have successfully run the software and benchmarked it.

Response to Reviewers

Reviewer #1

Comment: *This is an outstanding contribution that demonstrates truly impressive speed up of structural alignments based on the TM-score. The work is exceptionally well documented and carefully done and merits publication without revision. This will prove to be a very useful tool to the structural biology community.*

Reply: I wish to express my sincere gratitude to the reviewer for recognizing the significance of this work. This is deeply appreciated.

Comment: *Code is correct and very easy to follow.*

Reply: I am grateful for the reviewer's positive feedback on the clarity and correctness of the code.

Reviewer #2

Comment: *The manuscript submitted by Margelevicius entitled "GTalign: High-performance protein structure alignment, superposition, and search" describes a new algorithm for protein structure alignment, which, based on presented results, is on par with state-of-the-art structure alignment methods like TM-align in terms of accuracy while performing at much higher speed. While GTalign is an order of magnitude slower than Foldseek, based on the results presented, the accuracy of Foldseek is considerably lower. The presented tradeoff of GTalign in terms of accuracy and speed represents a major improvement in the field and thus, the presented results are expected to be of high interest to the field of computational structural biology.*

Reply: I sincerely appreciate the reviewer's recognition of the significance of the results and for their valuable feedback. I agree with all comments and addressed them carefully.

Major comments:

Comment: *While the manuscript is in principle well written in terms of English, it is very hard to understand conceptually the algorithmic approach and how this new algorithm is different compared to previously published approaches. At various places in the manuscript (abstract, introduction, results, and discussion) clarity of the manuscript could be improved by providing a proper introduction of the current state-of-the-art, conceptual layout of the novel approach, and discussion of the results with respect to previously published findings. The author should keep in mind that readers of Nature Communications come from a wider field of research but in its current state, the manuscript can only be understood by experts in dynamic programming, k-d tree data structure searches, etc.*

Reply: I have made significant revisions to the manuscript in line with the recommendations provided. Specifically, I have:

- Revised the abstract to provide more specificity;
- Expanded the introduction to include a discussion of the current state-of-the-art in protein structure alignment methods;
- Improved the outline of the algorithm in the Methods section (Sections 4.2–4.4);
- Enhanced the conceptual layout by adding Section 2.1 and expanding the description of Fig. 1a;
- Added a brief explanation of dynamic programming in Section 2.1 and introduced the k-d tree data structure in Section 4.3;
- Extended the discussion of the results;
- Conducted additional benchmarks and discussed the results in Section 2.

Please refer to Sections 1 through 4 to review these major changes.

Comment: *While the methods section, i.e. 4.2, is an attempt to more conceptually introduce the*

approach, it is not understandable because it lacks information on one side (i.e. “Assign secondary structure states to the structures based on the parameters optimized in [12].” -> so readers would need to now read upon reference 12 to understand what is done here?) while on the other hand suddenly provides a lot of technicalities that are not explained or are explained in the supplementary text (i.e. “Apply Algorithm 10...”). Furthermore, it is also not clear if all 10 steps presented in 4.2. are executed one after the other or represent alternative executions.

Reply: I have revised Section 4.2 and introduced two new subsections, 4.3 and 4.4, to provide a more detailed explanation of the GTalign algorithm.

In particular, Step 2 of the algorithm outline has been reformulated for clarity. I have omitted specific parameter values mentioned in reference 21, as these details are not essential for understanding the algorithm.

Assign secondary structure states to the structures at each residue in parallel. This assignment is determined by the coordinates of five residues centered around the residue under consideration, with distance cutoffs between residues optimized in [21].

Additionally, I have introduced Algorithm 10 in the second paragraph of Section 4.2 to provide context before outlining the GTalign algorithm:

Certain steps in the algorithm involve the alignment refinement procedure described in Algorithm 10 (RefineBestAlignments) specified in Supplementary Section S3. Algorithm 10 optimizes TM-scores and refines alignments by considering differently positioned alignment fragments of different lengths. It takes three parameters: the numbers of query and subject structures, where all possible pairs are processed in parallel, and a gap opening penalty for the COMER2 DP algorithm [27] to generate alignments that optimize TM-scores given the superpositions. The complete outline of the GTalign algorithm is provided below.

I have clarified in the first paragraph of Section 4.2 that all 10 steps outlined are executed sequentially:

Below, we outline the (sequential) algorithmic steps representing the actions performed on a pair of batch query and subject structures.

Comment: Fig 1a is supposed to illustrate the approach but lacks any conceptual detail and thus is not helpful to support understanding by the reader.

Reply: The purpose of Fig. 1a is to illustrate an essential aspect of the GTalign algorithm, which is further detailed in Section 4.4: the derivation of pairwise alignment through spatial matching in constant time. I have expanded the legend of Fig. 1a to clearly convey this significance.

Illustration of matching protein structures with $O(1)$ time complexity. GTalign explores

numerous superpositions in parallel. Upon obtaining a superposition, the alignment between the query protein (red) and the subject protein (blue) is generated using the subject protein's spatial index. This index allows for the independent retrieval of the nearest residue in the subject protein for each residue in the query protein, enabling parallel processing.

Comment: *The moderate performance of Foldseek in the various benchmark datasets is surprising, especially as it seems that essentially identical benchmark datasets were used here and in Kempen et al Nature Biotechnology 2023. In Kempen et al, Foldseek performed on par with methods like TM-align. It is clear that different approaches were used to evaluate performance of the various methods in Kempen et al and here, however, to improve clarity and comparability of the results presented here, it would be advisable to also perform evaluation of performances similar to approaches presented in Kempen et al, for example. If the author thinks that the evaluation procedure presented here is more accurate or relevant compared to evaluation procedures used elsewhere, then this should be laid out and supported with data.*

Reply: I have addressed both suggestions provided. First, I conducted additional benchmarking against the SCOPe dataset reference, following the approach employed by van Kempen et al. Generally, GTalign compares favorably to other tools, and these results are now detailed in Section 2.4 and 4.12, along with Supplementary Section S1.4.

Second, I have discussed the limitations of reference SCOPe-based evaluation in Section 2.5, with the discussion supported by data.

Please refer to these sections to review the new results and discussions.

Minor comments:

Comment: *References to figure panels are not in order (i.e. Fig 2 is referenced prior to Fig 1b) or missing (i.e. Fig S1).*

Reply: This has been corrected. I have carefully revised the text to ensure that all figures, panels, and Supplementary figures (with the exception of Fig. S9, for which Supplementary Section S2 is referenced) and tables are correctly referenced and presented in order.

Comment: *The different colors in panels of Fig 1 are very hard to distinguish. This is only possible upon close zoom in. Visualization could be improved either by labelling the different lines in the plot or by adding zoom-ins, for example.*

Reply: I have created Supplementary Figure S1 to depict zoomed-in sections of Fig. 1 and 2 from the main text. Additionally, I have included Supplementary Tables S1–S4, which present

the runtime and cumulative TM-score values, along with the number of top hits with TM-scores greater than 0.5, 0.6, 0.7, 0.8, and 0.9, for the data in Fig. 1 and 2. These data are now discussed in Section 2.2.

Comment: *It would be helpful to add to the legend of Fig 1 why different x and y scales were used in panels b–d. I assume this is because of the different dataset sizes but it would be good to clarify in the legend how the axis ranges were set.*

Reply: I have added the following text to the legend of Fig. 1 for clarification:

The axes scales in Panels b–d are chosen to accommodate the maximum values of the cumulative TM-score and the number of top hits with a TM-score ≥ 0.5 .

Comment: *Fig 1d. Why does Foldseek not extend to the right end of the x-axis range while the other tools do?*

Reply: I have included additional explanatory text in the legend of Fig. 1 to provide clarification:

The Foldseek curves appear truncated due to the total number of hits it produced.

Comment: *Page 5 lines 183–187 are unclear. What does the author mean by “spectral analysis of geometrical features in the frequency domain (as explored in our in-house research)”?*

Reply: I have revised the corresponding paragraph, now located in Section 2.2 as the penultimate paragraph, where I discuss using a single scoring scheme per protein pair. The particular sentence has been formulated as follows:

To address this, we explored the use of scores derived from spectral analysis of rotation-invariant two-dimensional representations of geometric features, such as angles and distances between residues, in the frequency (Fourier) domain.

Comment: *It is not clear what the author refers to when talking about the sequence prefiltering step for Foldseek. Also, can the author please better explain the two Foldseek options *talign-fast 0* and *1*?*

Reply: I have relocated and revised the discussion about sequence prefiltering to the penultimate paragraph in Section 2.2, directly following the explanation of decreased sensitivity resulting from

the prefiltering. This paragraph now reads as follows:

This phenomenon can be attributed to at least two factors. First, low sequence similarity does not necessarily correlate with low structural similarity, as demonstrated by the results. Second, the generation of accurate structural alignments using a single scoring scheme per protein pair may lack consistency. To address this, we explored the use of scores derived from spectral analysis of rotation-invariant two-dimensional representations of geometric features, such as angles and distances between residues, in the frequency (Fourier) domain. However, this approach demonstrated inconsistent results and requires further investigation. Despite these observations, leveraging prescreening in the sequence space can prove valuable for high-similarity searches.

I have revised the first paragraph of Section 2.2 to introduce the protein structure alignment tools and their variants, including the two Foldseek variants, that GTalign was benchmarked against. The description of the Foldseek variants has been improved with the following sentence from this paragraph:

We also evaluated the performance of TM-align's fast variant (option -fast) and Foldseek variants utilizing both fast (--talign-fast 1; FoldseekTM) and regular (--talign-fast 0) versions of TM-align for aligning protein structures that passed sequence similarity filters.

Reviewer #3

Comment: *Margelevicius et al. describe GTalign, a new method to compute pairwise protein structure alignments. The algorithm describes an iterative alignment algorithm utilizing spatial indices to improve search speed. The software is optimized for speed by comparing structures in batches and by utilizing GPU processing capabilities. A standout feature is how smoothly the software install went and that everything worked as expected with clear documentation.*

Reply: I am grateful to the reviewer for their review and positive evaluation of the software.

Comment: *The algorithmic idea of using spatial indices and the capability of batching multiple queries at once seem interesting and novel, but it would be great if they were discussed more in detail in the main manuscript in a broadly accessible way.*

Reply: I have introduced several new sections, 2.1, 4.3, and 4.4, to elaborate on the approach to superposition search using spatial indexing. Please refer to these sections to review them.

Regarding data processing in chunks, it is important to note that this algorithmic approach involves processing structure pairs within each chunk in parallel, applying the same set of algorithmic steps to each pair. These steps are outlined in the algorithms provided in Supplementary Section 3. For the low-level technical aspects of the data structures supporting parallel processing, extensive documentation is available in the source code.

Furthermore, the batch-oriented approach to processing data in chunks and the underlying data structures are similar to previously published techniques (Margelevicius, 2020, Bioinformatics). I have revised the first paragraph of Section 4.2 to reflect this:

The GTalign software takes as inputs query and subject (referred to as “reference” in the software) structure databases of arbitrary size. GTalign processes this data in chunks, aligning batches of query structures with batches of subject structures, both sorted by length, iteratively until all possible batch pairs are completed. A similar batch-oriented approach to processing large databases has been described previously [27].

Comment: *However, my main critique centers on the benchmarking of the software. First, regarding performance benchmarks: GTalign does not compare as favorably to TMalign as stated in the manuscript during my evaluation of the software on an A5000 GPU. Next, regarding sensitivity: The shown reference-free sensitivity benchmark results do not match the reference-based benchmark. This could be due to the reliance on a hard TMscore cut-off of 0.5, which is not a good discriminator whether a match is a true positive or false positive (at least according to community standards such as SCOP/CATH). The examples chosen to highlight the strength of the method in Supp. Fig. 2 are excellent examples why solely relying on a TMscore of > 0.5 is insufficient to detect similar folds reliably, since all shown example alignments are from different folds, with*

two even belonging to different architectures according to the SCOP hierarchy. Additionally, I have concerns regarding circularity of both conducting iterative TMscore optimization and using TMscore as an evaluation criterion.

Reply: I appreciate the reviewer’s critique and have thoroughly addressed all raised points while conducting additional benchmarks to address concerns. In particular, I have:

- Conducted additional analysis, which suggests the reviewer’s evaluation might have used the much slower CPU version of GTalign;
- Completed reference SCOPe-based benchmarking as requested;
- Discussed the limitations of benchmarking against the SCOPe reference extensively;
- Demonstrated the sensitivity of the tools in detecting structural similarity across different TM-score values;
- Conducted additional analysis to show significant structural similarities within and across SCOPe folds;
- Provided additional evidence showcasing significant structural similarity and matched topology between proteins in the examples;
- Evaluated alignments based on RMSD;
- Emphasized the use of TM-scores and RMSDs, the two most widely used measures of structural alignment accuracy, calculated with the independent established tool TM-align.

These points, along with responses to other comments, are discussed in detail below.

Major:

- *Sensitivity benchmark:*

Comment: *The benchmark has two major issues: The chosen measure of cumulative TMscore can become highly inflated by (over-)counting false positive alignments, which results in non-meaningful reported matches and harms the performance of methods that carefully reject false positive matches.*

Reply: This comment appears to diverge from the widely accepted understanding within the scientific community regarding the utility of TM-score as an established measure for quantifying structural similarity and the significance of structural similarity when TM-score > 0.5 (Xu and Zhang, 2010, Bioinformatics).

The reference-free benchmarks conducted in the manuscript were designed specifically to evaluate alignment accuracy across the entire TM-score significance range from 1.0 down to 0.5 and provide the progression of TM-scores across this range (also evidenced by Supplementary Table S2 and S4 included in the revision). An accurate tool that rejects false positive matches should demonstrate high alignment accuracy for true positives across this range (because alignments are sorted from the most accurate to the least accurate).

I assume the concerns regarding “false positives” and “non-meaningful reported matches” are triggered by alignments involving proteins with considerable differences in length. However, it is important to note that such alignments can indeed reflect significant structural similarities and reveal

shared topologies between shorter proteins and domains or segments within longer ones. This aspect is discussed in Section 2.5 (included in the revision), supported by relevant references, and exemplified in Fig. 6 (included in the revision) and Supplementary Figs. S2–S4.

To address these concerns more explicitly, Fig. 2 employs the TM-score normalized by the length of the query protein, effectively eliminating alignments involving substantially shorter subject proteins from evaluations (please see the discussion in Section 2.2).

Furthermore, the fact that the SCOPe40 2.08 dataset exclusively encompasses protein domains and that all queries of the Swiss-Prot dataset are relatively long proteins (>210 residues) further strengthens the relevance of these evaluations.

Finally, as articulated in the last paragraph of Section 1, GTalign’s principal objective is to facilitate the discovery of optimal protein superpositions and alignments, which can be sorted by various criteria using GTalign, enabling researchers to analyze proteins based on these detections:

GTaligns primary objective is to find an optimal spatial overlay for a given pair of structures and subsequently derive an alignment from it.

Therefore, the diverse and rigorous evaluations and benchmarking demonstrate exactly the opposite of the expressed concerns: efforts to focus on unbiased criteria to evaluate structural similarity.

Comment: *Additionally, a hard acceptance threshold of >0.5 TM-score allows alignments of proteins that do not share evolutionary history according to SCOP or CATH. SCOP and CATH classification on family, superfamily, and fold level are community standards to define evolutionary relationships and are commonly used to benchmark aligners. While the SCOP benchmarks shown in the manuscript use proteins contained within SCOP, they do not leverage the SCOP hierarchy.*

Reply: I performed reference SCOPe-based benchmarking following the approach used by van Kempen et al. (2023, Nat. Biotechnol.). GTalign demonstrates competitive performance, and these findings are now presented in Section 2.4 and 4.12, supplemented by Supplementary Section S1.4. Please refer to these sections for further details.

However, it is important to acknowledge the limitations of SCOPe-based evaluation in capturing the full spectrum of structural similarity and the advantages of TM-score-based reference-free evaluation. Here are my arguments:

A TM-score greater than 0.5 indicates significant structural similarity with high probability (Xu and Zhang, 2010, Bioinformatics). Therefore, the TM-score captures structural similarity, alignment accuracy, and sensitivity simultaneously. The TM-score progression (cumulative TM-score) for top-ranked alignments depicted in Fig. 1 and 2 renders a specific TM-score threshold irrelevant, reflecting the entire TM-score significance range. This is corroborated by Supplementary Table S2 and S4 (included in the revision), which detail the number of top hits at various TM-score thresholds. These results demonstrate the tools’ sensitivity in detecting structural similarity at different levels of structural similarity.

In contrast, SCOPe’s traditional emphasis on sequence similarity for classification (Fox et al., 2013,

Nucleic Acids Res.) implies that proteins highly divergent in length are unlikely to be classified into the same SCOPe fold despite shared topology and significant structural similarity.

Furthermore, the continuous nature of protein fold space challenges the discrete classification enforced by SCOPe, with significant structural and topological similarities extending across SCOPe folds and even classes. While some well-known cross-fold relationships are specified in Section 2.4 and in the paper by van Kempen et al., there are many more. Consequently, a correctly identified structural match is considered an error (false positive) if the identified proteins belong to different SCOPe folds. This issue undermines unbiased evaluation of structure alignment tools.

These points, supported by relevant references and references within these references, additional analysis, and examples, are discussed in Section 2.5 (included in the revision). Please refer to this section for further review.

Comment: Below is a table of average ROC1 (Fraction of TPs up to the first FP) values for family, superfamily, and fold following the SCOP benchmark proposed by Foldseek (van Kempen et al. 2023). From these numbers it can be seen that DALI performs best, followed by TAlign, GTalign, Foldseek-TM in a similar range, and Foldseek in last position. This stands in contrast with the results presented. Additionally, the table is annotated with the runtime of each method (see below).

Methods	Family	S.fam.	Fold	Speed in seconds
DALI	0.888076	0.575125	0.165703	123010
TAlign(fast)	0.892301	0.55768	0.163492	62240
Foldseek-TM	0.881055	0.547637	0.163978	1889
GTalign	0.88417	0.54846	0.157763	59748
Foldseek	0.86622	0.492773	0.108328	42

Reply: The reported sensitivity values may be inaccurate if calculated using the scripts from the van Kempen et al. study. (Details on the GTalign runtime are provided in response to a further comment below.)

There is a significant flaw in the scripts `bench.noselfhit.awk` and `bench.fdr.noselfhit.awk` found in the directory `scopbenchmark/scripts` of the <https://github.com/steineggerlab/foldseek-analysis> repository (downloaded on 02/24/2024).

The script `bench.noselfhit.awk` calculates the fractions of detected true positives (TPs) for each query at the SCOPe family, superfamily, and fold levels. However, these fractions are only calculated if the condition in line 24 is met:

```
if(id2fam[i] != "" && famCnt[id2fam[i]] > 1 && sfamCnt[id2sfam[i]] - famCnt[id2fam[i]] > 0 && foldCnt[id2fold[i]] - sfamCnt[id2sfam[i]] > 0){
```

This condition implies that queries are disregarded if they represent a family with only one member, a superfamily consisting of one family, or a fold consisting of one superfamily. Consequently, a significant fraction of queries is unjustifiably ignored. Notably, the calculation of TPs and their

counts in lines 18–21 adheres to the description in van Kempen et al.’s paper: “For family-level, superfamily-level and fold-level recognition, TPs were defined as same family, same superfamily and not same family and same fold and not same superfamily, respectively.”

The discrepancy arises after inserting between lines 30 and 31 the conditions for calculating TP fractions at the family, superfamily, and fold levels separately:

```
if(id2fam[i] != "" && famCnt[id2fam[i]] > 1) {...} ##... denotes calculations
if(id2fam[i] != "" && sfamCnt[id2sfam[i]] - famCnt[id2fam[i]] > 0) {...}
if(id2fam[i] != "" && foldCnt[id2fold[i]] - sfamCnt[id2sfam[i]] > 0) {...}
```

The number of queries processed (3566) in van Kempen et al.’s study for all tested methods accounts for only 40%, 41%, and 65% of valid queries at the family, superfamily, and fold levels, respectively.

The other script, `bench.fdr.noselfhit.awk`, calculates precision and recall values at the SCOPE family, superfamily, and fold levels. Similarly to the previous script, weighted TP values are only updated if the condition in line 39 is *not* satisfied:

```
(famCnt[id2fam[$1]] == 0 || sfamCnt[id2sfam[$1]] - famCnt[id2fam[$1]] == 0 ||
  foldCnt[id2fold[$1]] - sfamCnt[id2sfam[$1]] == 0){ next }
```

Consequently, the weighted TP values are not updated if a query represents a family with only one member, a superfamily consisting of one family, *or* a fold consisting of one superfamily. This leads to the unjustified exclusion of a majority of TPs, although their calculation adheres to van Kempen et al.’s description given above.

Differences become apparent after inserting before line 39 (after updating the weighted count of false positives) the conditions to update TP values separately at the family, superfamily, and fold levels:

```
(famCnt[id2fam[$1]] > 0 && id2fam[$1] == id2fam[$2]) {...} ##famCnt[id2fam[$1]]
  reduced by 1 already
((sfamCnt[id2sfam[$1]] - famCnt[id2fam[$1]] > 0) && (id2fam[$1] != id2fam[$2]) &&
  (id2sfam[$1] == id2sfam[$2])) {...} ##... denotes calculations
((foldCnt[id2fold[$1]] - sfamCnt[id2sfam[$1]] > 0) && (id2fam[$1] != id2fam[$2])
  && (id2sfam[$1] != id2sfam[$2]) && (id2fold[$1] == id2fold[$2])) {...}
```

In van Kempen et al.’s study, the numbers of TPs (unweighted) considered (44296, 136702, 379852) account for only 41%, 40%, and 65% of valid TPs at the family, superfamily, and fold levels, respectively, for TM-align and similarly for Dali.

Similar benchmarking was also employed by Holm (2019, Bioinformatics). The script `evaluate_ordered_lists.pl` in the benchmark data <http://ekhidna2.biocenter.helsinki.fi/dali/benchmark.tar.gz> counts TPs separately at the family, superfamily, and fold levels, in line with the proposed corrections.

In their paper, van Kempen et al. write (“Evaluation SCOPE benchmark”): “We evaluated only SCOPE members with at least one other family, superfamily and fold member.” It is unclear whether this sentence reflects the behavior programmed in their scripts and whether it represents the

```

struct          t hhhhhhhhhhhhhhhhhhhhh tt          ht ethh tthh t tthhhhhhhhhhhhhhhhh hh
Query: 90 -----PDMAFPRMNNMSFWLLPPSFLLLLASSMVEAGAGTGWTVYPLAGNLAHAGASVDLTIFSLHLAGVSSIL--GA 161
               +++
Refn.: 48 EIVEYAVKPLLAQSG-----PLD- 65
struct  hhhhhhhhhh t                                     th

struct  hhhhhhhh t th tthhhhhhhhhhhhhhhhhhhhhhhhhhhhhhh t hh tthhhhhhhhh
Query: 162 INFITTIINM-KPPAMSQYQTP-LFVWSVMITAVLLLSLPVLAAGITMLLTDRNLNTFFDPAGGGDPILYQHLFWFF- 238
               ++++++++ ++ + ++ ++++++++
Refn.: 66 -DIDVALRLIYAL--G---KMDKWLYADITHFSQYWHYLNEQDE-----TPG-F 107
struct  tthhhhhhhht tthhhhhhhhhhhhhhhhhhhhhhhhhhhhhhh t

struct  h hhhhhhh hhhhhhhhhhhhhhhhhhhhh t hhhhhhhhhhhhhhhhh hh ht tthhhhhhhhhhhhhhhhh
Query: 239 -G-HPEVYIL-ILPGFGMISHIVTYYSKKEPFYMGVMWAMMSIGFLGFIVWAHMFVGMVDVTRAYFTSATMIIAIP 315
               ++ +i + ++ i
Refn.: 108 ADDITWDFI-SNV--NS-I----- 122
struct  tthhhhhh h h h

struct  hhhhhhhhhhh t tthhhhhhhhhhhhhhhhhhhhhhhhhhhhh hhhhh ttthhhhhhhhhhhhhhhhhhhhhhhhhhhhh
Query: 316 TGVKVFSWLATLHGGNIKWSPAMMWALGFIFLFTVGGLTGIVLANSSLDIVLHDITYYVVAHFHYVLSMGAVFAIMGGFVH 395
               ++ ++L+++++++ + ++++++++V++ +++++A
Refn.: 123 ---TR-----NATLYDALKAMK-F-----V---WSEARFSGMVKT-ALTLAVTTTL-- 160
struct  hh hhhhhhhhh t tthhhhhhhhh hhhhhhhhhhh

struct  t hhht e tthhhhhhhhhhhhhhhhhhhhhhhhhhhhhhh ee e th hhhhhhhhhhhhhhhhhhhhhhhhhhhhh
Query: 396 WFPLFSGYTLNLTWAKIHFAIMFVGVNMTFFPQHFLGSLGMPRRYSYDPDAYTMWNTISSMGSFISLTAVMLMVFIIWEA 475
               +
Refn.: 161 K----- 161
struct  h

struct  hh eee e hh hh
Query: 476 FASKREVLTVDLTTTNLEW 494

Refn.: 162 -----E-LT 164
struct  h

```

```

Rotation [3,3] and translation [3,1] for Query:
  0.491805 -0.686111 -0.536078 297.037231
  0.306491  0.712693 -0.630977 -107.394035
  0.814979  0.146015  0.560793 -150.657990

```

This alignment reveals that all helices of d2hkta., except one (48-57), align with helices of d7coha., indicating matched topology. The spatial orientation of these helices matches those of d7coha.. The transformation matrix, and hence the superposition, produced by GTalign is equal (within numerical error) to that produced by TM-align using the GTalign alignment in FASTA format.

The TM-align output and the resulting matrix after issuing the command TAlign d7coha_.ent d2hkta_.ent -het 1 -I <GTalign_alignment_in_FASTA> -m <output_transformation_matrix> are as follows:

```

*****
* TM-align (Version 20220412): protein structure alignment *
* References: Y Zhang, J Skolnick. Nucl Acids Res 33, 2302-9 (2005) *
* Please email comments and suggestions to yangzhanglab@umich.edu *
*****

```

```
Name of Chain_1: ../structs/d7coha_.ent (to be superimposed onto Chain_2)
Name of Chain_2: ../structs/d2hkta_.ent
Length of Chain_1: 513 residues
Length of Chain_2: 164 residues

User-specified initial alignment: TM/Lali/rmsd = 0.54333, 145, 4.644
Aligned length= 145, RMSD= 4.64, Seq_ID=n_identical/n_aligned= 0.055
TM-score= 0.22271 (if normalized by length of Chain_1, i.e., LN=513, d0=8.03)
TM-score= 0.54333 (if normalized by length of Chain_2, i.e., LN=164, d0=4.77)
(You should use TM-score normalized by length of the reference structure)
```

```
("." denotes residue pairs of d < 5.0 Angstrom, "." denotes other aligned residues)
XFINRWLFSTNHKDIGTLYLLFGAWAGMVGTALESLLIRAELGQPGTLLGDDQIYNVVVTAHAFVMIFFMVMPIMIGGFGNWLVPLMIGA
-----PDMAPFRMNMSFWLLPPSFLLLASSMVEAGAGTGTWVYPLAGNLAHAGASVDLTIFSLHLAGVSSIL--GAINFI
TTIINM-KPPAMSQYQTP-LFVWSVMITAVLLLSPVLAAGITMLLTDRNLNTFFDPAGGGDPILYQHLPWFF--G-HPEVYIL-IL
PGFGMISHIVTYYSKKEPFYGMVWAMMSIGFLGFIVWAHMFVTGMDVDTRAYFTSATMIIAIPVGKVFWSWLATLHGNGIKWSPA
MMWALGFIFLFTVGGTLGIVLANSSLDIVLHDYYVVAHFHYVLSMGAVFAIMGGFVHWPLFSGYTLNDTWAKIHFAIMFVGVNMTFF
PQHFLGLSGMPRRYSYDPDAYTMWNTISSMGSFISLTAVMLMVFIWEAFASKREVLTVDLTTNLEWLNLCPPPYHTFEPTYVNL
      :.....: . . . . . : .. . . . . : . . . . .
      .. . . . .
.....: : . : : .. . . . . . . . . . . : . . . . . : . . . . .
: : . : : . : : .. . . . . : . . . . . : . . . . . :
: : . : : . : : .. . . . . : . . . . . : . . . . . :
: : . : : . : : .. . . . . : . . . . . : . . . . . :
      : .. . . . . :
      .. . . . .
-----MATLTEDDVLEQLD-A-----QDNLFSFMKTAHSI-LLQGIRQF-LPS-LF
VDNDEEIVEYAVKPLLAQSG-----PLD--DID
VALRLIYAL--G--KMDKWLYADITHFSQYWHYLNEQDE-----TPG-FADDITWDFI-SNV
--NS-I-----TR-----
-----NATLYDALKAMK-F-----V---WSEARFSGMVKT-ALTLAVTTTL--K-----
-----E-LT-----
```

Total CPU time is 0.04 seconds

```
----- The rotation matrix to rotate Chain_1 to Chain_2 -----
m          t[m]          u[m][0]          u[m][1]          u[m][2]
0    297.0358393252    0.4918098392    -0.6861065735    -0.5360791471
1    -107.3937124194    0.3064807539    0.7126954540    -0.6309791893
2    -150.6600768407    0.8149801406    0.1460238325    0.5607890965
```

```
Code for rotating Structure A from (x,y,z) to (X,Y,Z):
for(i=0; i<L; i++)
{
  X[i] = t[0] + u[0][0]*x[i] + u[0][1]*y[i] + u[0][2]*z[i];
  Y[i] = t[1] + u[1][0]*x[i] + u[1][1]*y[i] + u[1][2]*z[i];
  Z[i] = t[2] + u[2][0]*x[i] + u[2][1]*y[i] + u[2][2]*z[i];
}
```

The TM-align output remains the same even if the `-I` option is replaced with the `-i` option, which instructs TM-align to search for the optimal superposition starting with the one provided by the `-i` option.

These results suggest that TM-align does not enhance the alignment produced by GTalign. Furthermore, they validate that the examples for this domain pair, as well as others, are accurately depicted. They represent significant structural similarities with TM-scores ranging from 0.5 to 0.6,

which are prevalent within and across folds (Fig. 6).

There can be various reasons why these domains are classified into different folds by SCOPe. One factor is the discrete-continuous fold space duality, as discussed previously and in Section 2.5. Other reasons include low sequence identity and differences in subcellular location: d7coha_ belongs to the class of membrane proteins.

Indeed, these examples demonstrate that GTalign improves over other methods in detecting optimal superpositions, a feature of particular importance to structural biologists.

- *TMscore circularity:*

Comment: *Conducting iterative TMscore optimization and evaluating with TMscore as a cut-off in benchmarks seems circular and unfair. Choosing an independent measure (e.g. DALI Z-score) from the one being optimized should make the comparison much fairer.*

Reply: I conducted additional evaluation based on root-mean-squared deviation (RMSD), which is normalized by the number of aligned residue pairs and serves as a measure for local alignment accuracy evaluation. Optimizing alignments based on RMSD often results in short aligned fragments with low TM-scores. Therefore, RMSD represents an independent evaluation measure. For further details, please refer to Section 2.3.

The use of RMSD as a measure for quantifying protein alignment accuracy disregarding protein classifications was suggested by Kolodny et al. (2006, *Curr. Opin. Struct. Biol.*, doi:10.1016/j.sbi.2006.04.007, p. 396):

“In our opinion, instead of describing a protein as belonging to an existing or new fold, it would be more informative to report the value of a quantitative measure of structural similarity to one or more existing proteins. The similarity can then be quantified by the alignment’s properties (e.g. RMSD and length); given the alignment, these quantities are well defined and easily calculated.”

In the same paper, Kolodny et al. highlighted weaknesses of the Dali Z-score. The Dali Z-score is a statistical measure derived from internal raw scores. Despite the inability to calculate a Dali Z-score for a given alignment, the Dali Z-score does not directly quantify alignment accuracy but rather indicates the significance of the scores obtained. Thus, a higher Dali Z-score does not necessarily imply a more accurate alignment for the same pair of proteins.

Regarding the assertion of unfair benchmarking based on the TM-score, I must respectfully disagree. Here are the arguments:

First, no specific TM-score threshold was used to evaluate alignment accuracy. Instead, the progression of TM-scores (cumulative TM-score curves) reflects the rate of accurate alignments produced. The numbers of top hits at various TM-score thresholds can be located on these curves and are listed in Supplementary Table S2 and S4 (with Fig. 1 and 2 showing results for TM-score = 0.5).

Second, the TM-scores for all tools were calculated using an independent established tool, TM-align, ensuring fairness in benchmarking.

Third, the TM-score and RMSD are the two most widely used measures of structural alignment accuracy, with the TM-score being more common for proteins due to its avoidance of over-penalizing spatially unmatched residue pairs. These measures directly evaluate alignment accuracy, with higher spatial proximity resulting in a greater TM-score and lower RMSD.

Additionally, four out of six benchmarked tools (GTalign, TM-align, DeepAlign, and Foldseek) compute TM-scores, and three of them (GTalign, TM-align, and Foldseek-TM) optimize TM-scores, confirming the TM-score as a preferred objective function.

Finally, Yang Zhang, a co-author of TM-score and TM-align, co-authors (e.g., 2022, Nat. Methods; 2018, Bioinformatics, doi:10.1093/bioinformatics/btx828) and many others extensively use the TM-score for benchmarking their methods. Is their benchmarking unfair? Considering the widespread adoption of the TM-score, questioning the fairness of such benchmarking practices seems inappropriate.

- *Speed:*

Comment: *When benchmarking GTalign on an Nvidia A5000 GPU, I did not see speed advantages over TMalign, compared to paper’s claim of a one order of magnitude speedup over TMalign-fast. Here, TMalign (fast) was benchmarked on a 64-core machine. Foldseek seems to be also four orders of magnitude faster compared to GTalign (default) and not just one as claimed. How can this discrepancy be explained?*

I executed GTalign with the following command:
`gtalign -v -qrs=scop40.tar -rfs=scop40.tar -o test -s 0.`

Reply: The runtime performance indicated can be attributed to the utilization of the CPU version of GTalign, which is much slower. This assertion is supported by the results of the following analysis.

While I lack the ability to execute GTalign on an Nvidia RTX A5000 GPU, I replicated the experiment using GTalign with options `--qrs=scop40pdb.tar --rfs=scop40pdb.tar -o <output_directory>` (excluding verbosity `-v`; other options specified in Table 1 below) on a Tesla V100 GPU and a GeForce RTX 4090 Laptop GPU. It is worth noting that GTalign had previously been tested on Ampere architecture GPUs (A100), and the RTX A5000 is based on the Ampere architecture. (The SCOPe40 dataset was downloaded from <https://wwwuser.gwdg.de/~compbiol/foldseek>.)

The Tesla V100 GPU is outdated, and the GeForce RTX 4090 Laptop (Mobile) GPU is not as powerful as the RTX A5000 GPU (<https://technical.city/en/video/RTX-A5000-vs-GeForce-RTX-4090-mobile>).

The results, as presented in Table 1 below, demonstrate much faster runtimes compared to those indicated by the reviewer, even when utilizing less capable GPUs.

Moreover, the default parametrization of GTalign corresponds in accuracy to the standard TM-align version. (Please note that default option values may change in future versions.) As detailed in the manuscript, employing GTalign with the `--speed=13 --pre-score=0.4` options yields competitive

Table 1: GTalign runtimes in seconds on different GPUs

	1×Tesla V100	1×GeForce RTX 4090 Laptop
GTalign	14022.9	9377.6
GTalign -s 0	15137.4	10018.7
GTalign --speed=13 --pre-score=0.4	4001.4	2211.8
GTalign --speed=13 --pre-score=0.4 -s 0	4168.0	2306.3

accuracy at much faster runtimes. The significantly faster runtimes are also evident in Table 1. Additionally, it is worth noting that the `-s 0` option is unnecessary unless considering alignments with a TM-score < 0.5 .

For guidance on installing or running precompiled binaries of GTalign on CPUs and GPUs, comprehensive instructions are available in the GTalign documentation on the GitHub page. I encourage the reviewer to refer to these instructions before using GTalign.

- *Appropriate hardware configuration for benchmarking:*

Comment: *GTalign was benchmarked on three V100 GPUs and is compared against CPUs with 40 hardware threads in total. While it is difficult to fairly benchmark different hardware architectures in general, this evaluation does not seem fair from a performance/\$ perspective.*

The approximate cost of the two CPUs used should be 2000-3000\$ depending on the acquisition date, while the three V100 GPUs cost approximately 6000-30000\$ depending on when they were acquired. The fairest benchmark, given these prices, while keeping the same hardware configuration, would be to compare the two CPUs with 40 hardware threads vs. one V100 GPU.

Reply: I have addressed the reviewer’s request by including additional evaluations with GTalign runtimes on one along with two Tesla V100 GPUs in Supplementary Table S1.

However, there are several important reasons why utilizing all three Tesla V100 GPUs available on the system offers valuable insights for readers and does not compromise the integrity of fair benchmarking.

The evaluations were performed on a single server, utilizing all computational resources available, which reflects a typical research environment. One of GTalign’s features is its capability to utilize multiple GPUs, which is demonstrated in the benchmarking. Running GTalign on one, two, and three GPUs reveals its scalability properties.

Additionally, Supplementary Section S1.2 shows that a desktop-grade GeForce RTX 4090 GPU outperforms the three Tesla V100 GPUs. The price of the GeForce RTX 4090 GPU is \$1749 (<https://bestvaluegpu.com/history/new-and-used-rtx-4090-price-history-and-specs>).

The last paragraph of Section 2.2 summarizes these points:

GTalign offers additional computational advantages by providing the option to utilize

multiple GPUs for computation. This feature was effectively leveraged for processing the SCOPe40 2.08, PDB20, and Swiss-Prot datasets, where GTalign exploited the computational power of all three Tesla V100 GPUs available on the system. Supplementary Table S1 provides GTalign runtimes on one, two, and three GPUs, demonstrating scalability across all benchmarked parametrizations. Furthermore, the results presented in Supplementary Section S1.2 and Supplementary Table S5 unveil a noteworthy performance trend: A more recent desktop-grade GPU consistently outperforms the computational capabilities of the three server-grade V100 GPUs, effectively conveying GTalign's remarkable performance even when run on a single, relatively inexpensive GPU.

These results demonstrate that researchers can achieve the performance of GTalign demonstrated in the manuscript with a relatively inexpensive desktop computer equipped with one GPU. Therefore, I find no indication of any unfairness in the runtime evaluations conducted in the manuscript.

- *Explanation of algorithmic novelty*

Comment: *In its current form, the algorithmic novelty may not be easily comprehensible. Although the authors have included pseudocode in the supplementary materials, it remains challenging to understand. I suggest explaining, in particular, the constant-time comparison in detail, as this appears to be one of the most, if not the most important innovations of the work.*

Reply: I have updated Section 4.2 and added two additional subsections, 4.3 and 4.4, to provide a more in-depth overview and explanation of the GTalign algorithm. Please refer to these sections for review.

Comment: *I have successfully run the software and benchmarked it.*

Reply: I am grateful for the reviewer's feedback and time.

REVIEWER COMMENTS

Reviewer #2 (Remarks to the Author):

All my comments have been addressed.

Reviewer #3 (Remarks to the Author):

Upon reviewing the revised manuscript, I find that the primary concerns raised in my initial review remain unaddressed. In short, I have reevaluated the results of DALI, TAlign, Foldseek, and GAlign (see table below) and the performance differs from the report benchmark.

While the software works and generates good alignments, which in its own is quite useful as alternative aligner, the exaggerated claims about its speed and sensitivity in the manuscript do not align with my experience. I do have the feeling the issues might not be address in a next round of revision, so therefore I leave the decision to your experience.

Below are my comments of the revised manuscript:

Thank you for the efforts made to address the previous feedback on the manuscript. Despite these revisions, there remain several important issues:

- SCOP Benchmark Data: The updated data remains different from what I would expect.

Following is a reevaluation of the alignment files removing the mentioned exclusion conditions. The reevaluation resulted in the same trend as before: DALI outperforms the other methods, with TAlign, GAlign, and Foldseek-TM performing similarly, and Foldseek default scoring the lowest:

```
Foldseek 0.81664 0.375678 0.05029
Foldseek-TM 0.867467 0.417635 0.0802441
GAlign 0.880561 0.422145 0.0784656
TAlign 0.887643 0.427891 0.0819437
DALI 0.878771 0.455329 0.0785739
```

To proceed, I insist on a per-query comparison of each aligner's performance, such as a ROCX measure. This ensures that the Z-distribution of DALI is not unfairly penalized.

Figure 5: Exclusion of Rossmann folds: It is correct to remove the Rossmann fold (Figure 5a). However, it appears that all aligners except GAlign and TAlign suffer from issues when including Rossmann folds (Figure 5b). This observation is counterintuitive, especially since TAlign can easily align Rossmann folds, so I would expect a drop in performance.

- Cross-fold Hits (Figure 6): The cross-fold and cross-architecture hits are not convincingly demonstrated to be biologically meaningful alignments. I suspect that the issue here is the normalization of TM-scores by the length of the shorter protein.

In this case, a single helix could be aligned to any fold containing a single helix.

- Cumulative TMscore: I do not question the validity of TMscore. This is a question about the basic fairness of not comparing other methods on a given measure to a new method that was optimized against this measure. Many alternatives to TMscore exists and could be used instead e.g., LDDT iRMSD or GDT.

Comment:

- Software Speed on A5000 GPU: To confirm, I benchmarked the run time of an A5000 GPU and confirmed that it was utilizing the GPU. So it is still not clear how the performance discrepancy occurred. However, I did not re-assess with the newest version.

Response to Reviewers

Reviewer #2

Comment: *All my comments have been addressed.*

Reply: I sincerely thank the reviewer for their positive feedback. I greatly appreciate the valuable comments and the time taken by the reviewer to review the manuscript.

Reviewer #3

Comment: *Upon reviewing the revised manuscript, I find that the primary concerns raised in my initial review remain unaddressed. In short, I have reevaluated the results of DALI, TMalign, Foldseek, and GTalign (see table below) and the performance differs from the report benchmark.*

Reply: This comment is misleading for several reasons.

First, all concerns raised by the reviewer have been comprehensively addressed with additional analyses, as documented in the previous response-to-reviewer document.

Second, the reviewer appears to have used a flawed script in the first review, as indicated in a comment of the current review (“SCOP Benchmark Data... Following is a reevaluation”), to calculate average sensitivity values in the SCOPE-based evaluation.

Third, sensitivity plots were provided in Supplementary Figure S7 in the previous revision.

Fourth, in general, differences in absolute values are expected due to the use of different SCOPE datasets; however, the trend in average values remains consistent, as demonstrated below.

Finally, the average values provided by the reviewer appear to be still incorrect.

Comment: *While the software works and generates good alignments, which in its own is quite useful as alternative aligner, the exaggerated claims about its speed and sensitivity in the manuscript do not align with my experience. I do have the feeling the issues might not be address in a next round of revision, so therefore I leave the decision to your experience.*

Reply: I appreciate the reviewer’s feedback and would like to clarify that the claims regarding GTalign’s speed, accuracy, and sensitivity are firmly supported by data. The performance results presented in the manuscript are derived from rigorous benchmarking and comprehensive analyses. Furthermore, all analyses used to support these claims are transparently documented and available for verification.

Additionally, I want to assure the reviewer that all their comments have been thoroughly addressed in both the previous revision and the current one.

Comment: *Below are my comments of the revised manuscript:*

Thank you for the efforts made to address the previous feedback on the manuscript. Despite these revisions, there remain several important issues:

Reply: Every concern raised has been carefully considered and responded to with detailed expla-

nations and additional analyses where necessary.

Comment: - *SCOP Benchmark Data: The updated data remains different from what I would expect.*

Reply: Below, I demonstrate that the trend in average sensitivity values observed in the previous revision is consistent with the reviewer's expectations and remains unchanged in the current revision.

Comment: *Following is a reevaluation of the alignment files removing the mentioned exclusion conditions. The reevaluation resulted in the same trend as before: DALI outperforms the other methods, with TMalign, GTalign, and Foldseek-TM performing similarly, and Foldseek default scoring the lowest:*

```
Foldseek 0.81664 0.375678 0.05029
Foldseek-TM 0.867467 0.417635 0.0802441
GTalign 0.880561 0.422145 0.0784656
TMalign 0.887643 0.427891 0.0819437
DALI 0.878771 0.455329 0.0785739
```

To proceed, I insist on a per-query comparison of each aligner's performance, such as a ROCX measure. This ensures that the Z-distribution of DALI is not unfairly penalized.

Reply: First, I confirmed that the reviewer is referring to the average ROC1 ("Fraction of TPs up to the first FP") values by recalculating these values using the original flawed script `bench.noselfhit.awk` and comparing the output with the values provided by the reviewer in their previous review.

Next, I corrected the `bench.noselfhit.awk` script and reevaluated the average sensitivity (ROC1) values on the SCOPe40 dataset from <https://wwwuser.gwdg.de/~compbiol/foldseek> using the corrected `bench.noselfhit.awk` script and the `GetRecall1` function in the `commands-test-scope20840-fr2.sh` script accompanying this manuscript.

Below, I compare and discuss the results.

The blocks of corrections introduced in the `bench.noselfhit.awk` script are commented and delineated by the opening and end delimiters `##{MM: and ##}`. The corrected `bench.noselfhit.awk` script is provided below:

```
#!/usr/bin/mawk -f
BEGIN{OFS="\t";
  ##{MM: adjust the heading for average values:
  ##print "NAME", "SCOP", "FAM", "SFAM", "FOLD", "FP", "FAMCNT", "SFAMCNT", "FOLDCNT";
  print "FAM", "SFAM", "FOLD", "FAMSUM", "SFAMSUM", "FOLDSUM", "FAMCNT", "SFAMCNT", "FOLDCNT";
  ##}}
}
FNR==NR{
  id2fam[$1]=$2;
  famCnt[$2]++;
}
```

```

gsub(/\. [0-9]+\$/, "", $2);
id2sfam[$1]=$2;
sfamCnt[$2]++;
gsub(/\. [0-9]+\$/, "", $2);
id2fold[$1]=$2;
foldCnt[$2]++;
next }
!($1 in id2fam) {next}
!($2 in id2fam) {foundFp[$1]++; next}
$1 == $2 {next} # skip self hit
foundFp[$1] < 1 && id2fold[$1] != id2fold[$2] {foundFp[$1]++; next}
foundFp[$1] < 1 && id2fam[$1] == id2fam[$2] { foundFam[$1]++; next }
foundFp[$1] < 1 && id2fam[$1] != id2fam[$2] && id2sfam[$1] == id2sfam[$2] { foundSFam[$1]++; next }
foundFp[$1] < 1 && id2fam[$1] != id2fam[$2] && id2sfam[$1] != id2sfam[$2] &&
  id2fold[$1] == id2fold[$2] {
  foundFold[$1]++; next
}
}
END{
##{MM: initialize variables:
famVal = 0; sfamVal = 0; foldVal = 0;
nqrsfam = 0; nqrssfam = 0; nqrsfold = 0;
##}
for(i in id2fam){
##{MM: comment out this entire block:
##if(id2fam[i] != "" && famCnt[id2fam[i]] > 1 &&
##  sfamCnt[id2sfam[i]] - famCnt[id2fam[i]] > 0 &&
##  foldCnt[id2fold[i]] - sfamCnt[id2sfam[i]] > 0) {
##  famVal=foundFam[i]/(famCnt[id2fam[i]] - 1);
##  sfamVal=foundSFam[i]/(sfamCnt[id2sfam[i]] - (famCnt[id2fam[i]] - 1));
##  foldVal=foundFold[i]/(foldCnt[id2fold[i]] - (sfamCnt[id2sfam[i]] - 1));
##  fpCnt = (foundFp[i] == "") ? 0 : foundFp[i];
##  print i, id2fam[i], famVal, sfamVal, foldVal,
##      fpCnt, famCnt[id2fam[i]], sfamCnt[id2sfam[i]], foldCnt[id2fold[i]];
##}
##}
##{MM: introduce corrections: calculate TP fractions at all levels separately:
if(id2fam[i] != "" && famCnt[id2fam[i]] > 1) {
  famVal = famVal + foundFam[i]/(famCnt[id2fam[i]] - 1); nqrsfam++;
}
if(id2fam[i] != "" && sfamCnt[id2sfam[i]] - famCnt[id2fam[i]] > 0) {
  ##{MM: no subtraction of 1 from famCnt[id2fam[i]]: different families considered}
  sfamVal = sfamVal + foundSFam[i]/(sfamCnt[id2sfam[i]] - famCnt[id2fam[i]]); nqrssfam++;
}
if(id2fam[i] != "" && foldCnt[id2fold[i]] - sfamCnt[id2sfam[i]] > 0) {
  ##{MM: no subtraction of 1 from sfamCnt[id2sfam[i]]: different superfamilies considered}
  foldVal = foldVal + foundFold[i]/(foldCnt[id2fold[i]] - sfamCnt[id2sfam[i]]); nqrsfold++;
}
##}
}
##{MM: print average values, sums, and effective numbers of queries:
print famVal/nqrsfam, sfamVal/nqrssfam, foldVal/nqrsfold,
      famVal, sfamVal, foldVal, nqrsfam, nqrssfam, nqrsfold;
##}
}

```

The original and corrected scripts accept sorted data provided by van Kempen et al. (2023, Nat. Biotechnol.). However, the Foldseek data was only partially sorted. The example below shows a

beginning excerpt of the >58M lines, with unordered E -values highlighted in boldface:

d1a99a_.pdb	d1i6aa_.pdb	0.106	282	160	18	1	272	4	203	5.647E-04	92
d1a99a_.pdb	d2c21a1.pdb	0.084	367	192	22	6	340	3	257	1.743E-04	90
d1a99a_.pdb	d3k4ua_.pdb	0.116	293	152	17	1	272	2	208	4.514E-04	88
d1a99a_.pdb	d3fasa_.pdb	0.106	328	187	20	1	291	1	259	9.344E-04	88
d1a99a_.pdb	d1wdna_.pdb	0.117	289	146	19	4	272	2	201	6.315E-04	86
d1a99a_.pdb	d2esna2.pdb	0.084	285	163	19	2	273	8	207	3.027E-03	83
d1a99a_.pdb	d2fyia1.pdb	0.112	285	153	20	3	273	7	205	8.355E-04	83
d1a99a_.pdb	d3i6va_.pdb	0.080	284	155	17	4	272	2	194	2.420E-03	78
d1a99a_.pdb	d1utha_.pdb	0.117	282	154	20	2	272	12	209	1.729E-03	77
d1a99a_.pdb	d3tqla_.pdb	0.125	296	147	21	1	272	1	208	1.178E-04	76
d1a99a_.pdb	d1h3da1.pdb	0.114	306	155	15	2	278	1	219	1.383E-03	71
d1a99a_.pdb	d2ozza1.pdb	0.100	300	152	20	1	279	12	214	2.420E-03	70
d1a99a_.pdb	d1xs5a_.pdb	0.100	299	159	21	2	272	3	219	3.201E-03	70
d1a99a_.pdb	d1ixha_.pdb	0.103	368	213	23	1	307	1	312	1.635E-03	70
d1a99a_.pdb	d1o63a_.pdb	0.089	292	157	17	4	279	1	199	4.004E-03	69
d1a99a_.pdb	d1ryoa_.pdb	0.120	307	191	17	2	270	1	266	1.383E-03	68
d1a99a_.pdb	d1nh8a1.pdb	0.072	291	159	18	4	273	2	202	1.226E-02	63
d1a99a_.pdb	d2i6ea1.pdb	0.105	371	187	24	5	339	2	263	1.105E-03	61
d1a99a_.pdb	d1r91a_.pdb	0.114	385	201	27	3	335	9	305	6.265E-03	61
d1a99a_.pdb	d1pc3a_.pdb	0.100	410	206	26	3	338	11	331	4.478E-03	61
d1a99a_.pdb	d1ve4a1.pdb	0.082	292	150	20	2	273	3	196	9.803E-03	60
d1a99a_.pdb	d1twya_.pdb	0.090	322	167	19	2	279	1	240	1.297E-02	60
d1a99a_.pdb	d2nxoa1.pdb	0.145	372	168	29	1	322	1	272	2.862E-03	60
d1a99a_.pdb	d2a5ea1.pdb	0.082	329	172	19	2	271	1	258	3.766E-02	53
d1a99a_.pdb	d1z7me1.pdb	0.067	298	150	17	4	273	2	199	3.551E-02	50
d1a99a_.pdb	d1pb7a_.pdb	0.076	338	166	20	2	270	1	261	4.968E-02	49
d1a99a_.pdb	d3delb_.pdb	0.115	312	156	24	1	284	1	220	1.622E-02	48
d1a99a_.pdb	d11cfa2.pdb	0.129	324	175	23	4	271	10	282	3.972E-02	47
d1a99a_.pdb	d2fria2.pdb	0.176	51	35	3	2	46	25	74	3.726E-01	36
d1a99a_.pdb	d1npya2.pdb	0.154	71	49	5	2	66	7	72	2.816E-01	35
d1a99a_.pdb	d1pdaa1.pdb	0.093	300	139	21	1	263	1	204	2.013E-01	35
d1a99a_.pdb	d2qwa1.pdb	0.110	272	120	18	5	253	2	174	7.351E-02	32
d1a99a_.pdb	d1r8ja2.pdb	0.132	53	43	2	1	52	2	52	1.427E+00	31
d1a99a_.pdb	d3b1ga_.pdb	0.095	188	95	11	104	269	116	250	1.216E-01	31
d1a99a_.pdb	d3g63a_.pdb	0.104	430	219	30	1	322	1	372	2.381E-01	31
d1a99a_.pdb	d1edza2.pdb	0.114	61	44	3	2	54	35	93	7.712E-01	30
d1a99a_.pdb	d1wu7a1.pdb	0.200	75	46	6	1	69	3	69	5.512E-01	28
d1a99a_.pdb	d1y5ea1.pdb	0.161	99	66	6	1	82	2	100	1.207E+00	28
d1a99a_.pdb	d1xoca1.pdb	0.091	362	146	24	27	266	198	498	2.129E-01	28
d1a99a_.pdb	d1vi2a2.pdb	0.129	62	46	3	2	57	4	63	1.141E+00	27

This partially sorted Foldseek data leads to heightened but inaccurate results at all levels. Therefore, before calculating average sensitivity values, I sorted the Foldseek data by E -value and removed the .pdb extension from the domain identifiers in both the Foldseek and Foldseek-TM data.

Regarding GTalign, since the manuscript provides evaluation results for GTalign based on the harmonic mean of the secondary TM-scores (2TM-scores), I executed the default version of GTalign (`--speed=9`) with the `--2tm-score` option and used the harmonic mean of the 2TM-scores to sort its alignments, as specified in the manuscript.

The average sensitivity values calculated using the corrected `bench.noselfhit.awk` script for the data provided by van Kempen et al. and GTalign are specified in Table 1.

Table 1: Average sensitivity (ROC1) values calculated using the corrected `bench.noselfhit.awk` script. S_{family} , $S_{\text{s.family}}$, and S_{fold} represent the sums of the fractions of TPs before encountering the first FP at the family, superfamily, and fold levels, respectively. N_{family} , $N_{\text{s.family}}$, and N_{fold} indicate the effective numbers of queries at the family, superfamily, and fold levels.

	Family	Superfamily	Fold	S_{family}	$S_{\text{s.family}}$	S_{fold}	N_{family}	$N_{\text{s.family}}$	N_{fold}
Foldseek	0.829829	0.500536	0.079456	7447.72	4322.63	436.057	8975	8636	5488
Foldseek-TM	0.867564	0.595820	0.174480	7786.38	5145.50	957.549	8975	8636	5488
GTalign	0.880348	0.602599	0.175123	7901.12	5204.05	961.076	8975	8636	5488
TMalignFast	0.886929	0.606912	0.175459	7960.19	5241.29	962.919	8975	8636	5488
DALI	0.878722	0.644613	0.169542	7886.53	5566.88	930.444	8975	8636	5488

Table 2 presents the average sensitivity values calculated for the same data using the `GetRecall11` function in the `commands-test-scope20840-fr2.sh` script of this manuscript, additionally instructed to print average values. (The `GetRecall11` function always sorts data and provides the same results irrespective of whether input data are sorted.)

Table 2: Average sensitivity (ROC1) values calculated using the `GetRecall11` function of this manuscript.

	Family	Superfamily	Fold	S_{family}	$S_{\text{s.family}}$	S_{fold}	N_{family}	$N_{\text{s.family}}$	N_{fold}
Foldseek	0.829829	0.500536	0.0794564	7447.72	4322.63	436.057	8975	8636	5488
Foldseek-TM	0.867564	0.595820	0.1744804	7786.38	5145.50	957.549	8975	8636	5488
GTalign	0.880348	0.602599	0.1751231	7901.12	5204.05	961.076	8975	8636	5488
TMalignFast	0.886929	0.606912	0.1754590	7960.19	5241.29	962.919	8975	8636	5488
DALI	0.878722	0.644613	0.1695415	7886.53	5566.88	930.444	8975	8636	5488

The results in Table 1 and 2 show that the corrected `bench.noselfhit.awk` script and the `GetRecall11` function of this manuscript produce identical average values and related statistics.

These results also show that the average values provided by the reviewer are only similar in absolute value at the family level but incorrect at the superfamily and fold levels. However, the trend in ranking by the average sensitivity values is consistent at the family, superfamily, and fold levels.

Now, I turn to the results presented in Supplementary Figure S7 and Supplementary Table S8 (added in this revision) of the manuscript.

Supplementary Figure S7 has been revised to plot sensitivity values against the effective fraction of queries at the family, superfamily, and fold levels. In the previous revision, Supplementary Figure S7 plotted sensitivity values against the fraction of queries calculated from the total number of queries, 2045. (Differences between plotting these graphs using the current and previous scripts can be inspected using the Linux command `diff plot_TMscores_scope20840-fr2.sh plot_TMscores_scope20840-fr.sh`. Differences in the `GetRecall11` function can be identified using the command `diff commands-test-scope20840-fr2.sh commands-test-scope20840-fr.sh`.)

Please note that the relative difference between the curves of the tested tools remains the same in the previous version of Supplementary Figure S7 and the current one because the sensitivity values of all tools are normalized by the same number (the total number of queries previously and the effective number of queries in the current revision).

The ranking of the tools by average sensitivity values in Supplementary Table S8 aligns with the reviewer’s expectations based on their table of average ROC1 values, except for the relative performance between Foldseek-TM and DALI at the fold level.

In summary, there are several important points to note. First, the sensitivity (ROC1) analyses, even when conducted on different SCOPe datasets, produce consistent results that align with the reviewer’s expectations. These results are also consistent with the previous version of the sensitivity plot (Supplementary Figure S7).

Second, the difference in average sensitivity values between GTalign and the other top performers in this evaluation is statistically insignificant at the family, superfamily, and fold levels (Supplementary

Table S8).

Comment: *Figure 5: Exclusion of Rossmann folds: It is correct to remove the Rossmann fold (Figure 5a). However, it appears that all aligners except GTalign and TMalign suffer from issues when including Rossmann folds (Figure 5b). This observation is counterintuitive, especially since TMalign can easily align Rossmann folds, so I would expect a drop in performance.*

Reply: It is not accurate to state that GTalign and TM-align do not experience a drop in precision when pairs from different Rossmann-like or beta-propeller folds are considered errors (false positives).

In fact, all tools, including GTalign and TM-align, experience a performance drop when these pairs are considered errors, compared to when they are ignored. This phenomenon is reflected in the area under the weighted precision-recall curves (AUPRC), as shown in Supplementary Table S7.

Please refer to Supplementary Table S7 for details.

When rounding to three decimal places, Supplementary Table S7 shows that the AUPRC values for FATCAT are equal at the family and fold levels. However, differences become apparent with increased precision. The AUPRC values for FATCAT at the family, superfamily, and fold levels are as follows:

Cross-fold relationships ignored: 0.603283, 0.365756, 0.012297

No exceptions: 0.602837, 0.365402, 0.012254

Comment: - *Cross-fold Hits (Figure 6): The cross-fold and cross-architecture hits are not convincingly demonstrated to be biologically meaningful alignments. I suspect that the issue here is the normalization of TM-scores by the length of the shorter protein. In this case, a single helix could be aligned to any fold containing a single helix.*

Reply: I have included Supplementary Figure S8 in this revision, which plots the distribution of TM-scores normalized by the query length.

Please refer to Supplementary Figure S8 for details.

Comment: - *Cumulative TMscore: I do not question the validity of TMscore. This is a question about the basic fairness of not comparing other methods on a given measure to a new method that was optimized against this measure. Many alternatives to TMscore exists and could be used instead e.g., LDDT iRMSD or GDT.*

Reply: This comment was thoroughly addressed in the previous revision. Please refer to page 15 of the previous response-to-reviewers document for the detailed response.

In the previous revision, I evaluated alignment accuracy based on RMSD, an independent measure, as requested by the reviewer.

To further address this comment in the current revision, I have additionally evaluated alignment accuracy based on global distance test (GDT) scores (GDT_TS), as suggested by the reviewer.

Please refer to Figure 4 and Supplementary Table S6 for these results.

Regarding other suggested measures, iRMSD, which measures the RMSD of inter-chain interface residues, is not suitable for the single-chain structure analyses performed in this manuscript. The local distance difference test (LDDT) is used to evaluate structural model quality with respect to an experimental structure. I am not aware of any tool that calculates LDDT scores for given alignments between two different protein structures.

As a final remark, protein structure alignment methods are developed to infer biological insights, and the TM-score is perhaps the most widely adopted measure for protein structure similarity. If there were a more appropriate measure, GTalign would optimize it.

Comment: *Comment:*

- *Software Speed on A5000 GPU: To confirm, I benchmarked the run time of an A5000 GPU and confirmed that it was utilizing the GPU. So it is still not clear how the performance discrepancy occurred. However, I did not re-assess with the newest version.*

Reply: To dispel any doubts, I have conducted additional GTalign tests on an Nvidia RTX A5000 GPU on the cloud (<https://vast.ai>, <https://cloud.vast.ai>).

I downloaded the GTalign version 0.14.0 software package, the same version the reviewer used, from https://github.com/minmarg/gtalign_alpha/archive/refs/tags/v0.14.0-alpha-doi.tar.gz and ran GTalign with the options `--qrs=scop40pdb.tar --rfs=scop40pdb.tar -o <output_directory>` (other options specified in Table 3 below) on a vast.ai compute instance with an A5000 GPU. These are the same options indicated on page 16 of the previous response-to-reviewers document in response to the reviewer's comment on the same issue. Please note that the CUDA version (≥ 10.1) has no significant impact on GTalign runtimes.

The `nvidia-smi` output showing the full GPU utilization in the vast.ai instance is provided below:

```
Sun Jun 16 10:02:13 2024
+-----+
| NVIDIA-SMI 535.171.04                Driver Version: 535.171.04   CUDA Version: 12.2   |
+-----+-----+-----+-----+-----+-----+
| GPU  Name                               Persistence-M | Bus-Id        Disp.A | Volatile Uncorr. ECC |
| Fan  Temp   Perf              Pwr:Usage/Cap |      Memory-Usage | GPU-Util  Compute M. |
|                                           MIG M.         |                      |                      |
+-----+-----+-----+-----+-----+-----+
|   0   NVIDIA RTX A5000                   On          | 00000000:01:00.0 Off  |                     Off | |
| 61%   73C    P2              187W / 190W | 23984MiB / 24564MiB |    100%    Default  |
|                                           |                      |                      | N/A |
+-----+-----+-----+-----+-----+-----+

```

```

+-----+
| Processes:                                     |
| GPU  GI  CI          PID  Type  Process name                               |
|      ID ID          |          |          |          GPU Memory |
|      ID ID          |          |          |          Usage      |
+-----+

```

While the `nvidia-smi` program does not list processes in virtual machines, the GTalign process running is shown (command lines truncated by `ps`) using the command `ps f -o user,pid,pcpu,pmem,command`:

```

USER          PID %CPU %MEM COMMAND
root          1094  0.0  0.0 -bash
root          1479  0.0  0.0 \_ ps f -o user,pid,pcpu,pmem,command
root          1079  0.0  0.0 -bash
root          1091  0.0  0.0 \_ tmux new-session -s ssh_tmux
root          1183  0.0  0.0 bash -c (time gtalign_alpha-0.14.0-alpha-doi/Linux_installer_GPU/bin/
gtalign --qrs=scop40pdb.tar
root          1184  169  0.4 \_ gtalign_alpha-0.14.0-alpha-doi/Linux_installer_GPU/bin/gtalign --qrs=
scop40pdb.tar --rfs=sco

```

The results are presented in Table 3 below:

Table 3: GTalign runtimes in seconds on an RTX A5000 GPU

	1×RTX A5000
GTalign	11968.2
GTalign -s 0	12762.8
GTalign --speed=13 --pre-score=0.4	2634.1
GTalign --speed=13 --pre-score=0.4 -s 0	2737.9

The results in Table 3 demonstrate that GTalign works as intended and runs properly on the A5000 GPU with much faster runtimes than those reported by the reviewer.

These results also highlight the inaccuracies in the runtimes provided by the reviewer in their previous review (page 10 of the previous response-to-reviewers document).

REVIEWERS' COMMENTS

Reviewer #3 (Remarks to the Author):

I would like to thank the authors for addressing my comments.
Two points remain, which should not require my further review.

- Table 1 follows the trend I would expect with one exception. Foldseek's hits should not be sorted by E-value; please retain the default output order, which sorts by structural bit score.
- Additionally, adding runtime to Table 1 would help the read to see this numbers in perspective.

Reviewer #3 (Remarks on code availability):

Code compiles and runs as expected,

Response to Reviewers

Reviewer #3

Comment: *I would like to thank the authors for addressing my comments.*

Reply: I appreciate the reviewer’s feedback and comments.

Comment: *Two points remain, which should not require my further review.*

Reply: Both points are addressed below.

Comment: - *Table 1 follows the trend I would expect with one exception. Foldseek’s hits should not be sorted by E-value; please retain the default output order, which sorts by structural bit score.*

Reply: This and the following comment indicate that the reviewer refers to Table 1 from the previous (second revision) response-to-reviewers document. The manuscript itself does not contain Table 1, and Supplementary Table S1, which provides runtimes, represents the results of an alignment accuracy and structural similarity benchmark.

I have updated Table 1 (provided below), which now shows Foldseek’s results obtained using its data file in the original order as provided by van Kempen et al. (2023, Nat. Biotechnol.).

Table 1: Average sensitivity (ROC1) values calculated using the corrected `bench.noselfhit.awk` script. S_{family} , $S_{\text{s.family}}$, and S_{fold} represent the sums of the fractions of true positives (TPs) before encountering the first false positive (FP) at the family, superfamily, and fold levels, respectively. N_{family} , $N_{\text{s.family}}$, and N_{fold} indicate the effective numbers of queries at the family, superfamily, and fold levels. t represents runtime in seconds.

	Family	Superfamily	Fold	S_{family}	$S_{\text{s.family}}$	S_{fold}	N_{family}	$N_{\text{s.family}}$	N_{fold}	t (s)
Foldseek	0.857422	0.554974	0.110351	7695.36	4792.76	605.605	8975	8636	5488	42
Foldseek-TM	0.867564	0.595820	0.174480	7786.38	5145.50	957.549	8975	8636	5488	1889
GTalign	0.880348	0.602599	0.175123	7901.12	5204.05	961.076	8975	8636	5488	5571
TMalignFast	0.886929	0.606912	0.175459	7960.19	5241.29	962.919	8975	8636	5488	62240
DALI	0.878722	0.644613	0.169542	7886.53	5566.88	930.444	8975	8636	5488	123010

Foldseek alignments are sorted by E -value (the section “SCOPE-based evaluation”) to produce Figure 5, Supplementary Figure S7, and Supplementary Tables S7 and S8 in the manuscript because Foldseek does not output structural bit scores, and there is no way to calculate them from output data for sorting the full set of produced alignments. According to the Foldseek manual (and paper), the structural bit score is defined as `bits*sqrt(alnlddt*alntmscore)`, but `alnlddt` cannot be

output.

Comment: - *Additionally, adding runtime to Table 1 would help the read to see this numbers in perspective.*

Reply: Runtimes have been added to Table 1.

The runtimes for all tools except GTalign in Table 1 correspond to those specified by the reviewer in the first review (page 10 of the first response-to-reviewers document) and match those reported by van Kempen et al. (2023, Nat. Biotechnol.).

These runtimes were obtained on different machines: all tools except GTalign were run on 64 CPU cores, while GTalign was run on three V100 GPUs. As indicated in the manuscript, collectively running GTalign on three V100 GPUs is inferior in performance to running it on a single GeForce RTX 4090 GPU.

Additionally, the GTalign results in Table 1 correspond to the current default version (`--speed=9`), which roughly matches in alignment accuracy the regular TM-align version. Other GTalign parametrizations produce similar results with much faster runtimes.

Comment: *Code compiles and runs as expected.*

Reply: I appreciate the reviewer's feedback and time.